# The genomic landscape of cholangiocarcinoma reveals the disruption of post-transcriptional modifiers

Yaodong Zhang [1,6], Zijian Ma [2,6], Changxian Li[1,6], Cheng Wang [2,3,4✉], Wangjie Jiang[1], Jiang Chang[1], Sheng Han[1], Zefa Lu[1], Zicheng Shao[1], Yirui Wang[1], Hongwei Wang[1], Chenyu Jiao[1], Dong Wang[1], Xiaofeng Wu[1], Hongbing Shen [2,4], Xuehao Wang [1], Zhibin Hu [2,4,5✉] & Xiangcheng Li [1✉]

Molecular variation between geographical populations and subtypes indicate potential genomic heterogeneity and novel genomic features within CCA. Here, we analyze exome-sequencing data of 87 perihilar cholangiocarcinoma (pCCA) and 261 intrahepatic cholangiocarcinoma (iCCA) cases from 3 Asian centers (including 43 pCCAs and 24 iCCAs from our center). iCCA tumours demonstrate a higher tumor mutation burden and copy number alteration burden (CNAB) than pCCA tumours, and high CNAB indicates a poorer pCCA prognosis. We identify 12 significantly mutated genes and 5 focal CNA regions, and demonstrate common mutations in post-transcriptional modification-related potential driver genes METTL14 and RBM10 in pCCA tumours. Finally we demonstrate the tumour-suppressive role of METTL14, a major RNA N6-adenosine methyltransferase (m6A), and illustrate that its loss-of-function mutation R298H may act through m6A modification on potential driver gene MACF1. Our results may be valuable for better understanding of how post-transcriptional modification can affect CCA development, and highlight both similarities and differences between pCCA and iCCA.

---

[1] Hepatobiliary Center, The First Affiliated Hospital of Nanjing Medical University, Key Laboratory of Liver Transplantation, Chinese Academy of Medical Sciences, NHC Key Laboratory of Liver Transplantation, Nanjing 210029, China. [2] Department of Epidemiology, Center for Global Health, School of Public Health, Nanjing Medical University, Nanjing, China. [3] Department of Bioinformatics, School of Biomedical Engineering and Informatics, Nanjing Medical University, Nanjing, China. [4] Jiangsu Key Lab of Cancer Biomarkers, Prevention and Treatment, Collaborative Innovation Center for Cancer Personalized Medicine, Nanjing Medical University, Nanjing, China. [5] State Key Laboratory of Reproductive Medicine, Center for Global Health, Nanjing Medical University, Nanjing, China. [6]These authors contributed equally: Yaodong Zhang, Zijian Ma, Changxian Li. ✉email: cheng_wang29@njmu.edu.cn; zhibin_hu@njmu.edu.cn; drxcli@njmu.edu.cn

Cholangiocarcinoma (CCA) is a malignant primary hepatobiliary disease originating from every point of the biliary tree, from the canals of Hering to the main bile duct[1]. In recent decades, the incidence and mortality of CCA have increased globally, especially in East Asia[2,3]. In China, the incidence of CCA reached 6–7.55 per 100,000 individuals[4,5]. The mortality of CCA is almost equal to its incidence due to resistance to common treatments and poor prognosis[6]. The overall 5-year survival rate is only 10%[7]. Even for the patients with CCA who are amenable to surgical resection, and the corresponding five-year survival rate is only 25–35%[8], and high recurrence rates of 50 to 60% persist upon diagnosis, with no effective therapy.

CCA is regarded as a group of different diseases that are further divided into intrahepatic CCA (iCCA), perihilar CCA (pCCA), and distal CCA (dCCA) based on anatomical location[9]. pCCA is the most frequent subtype, accounting for 50–60% of all CCA tumors[9]. In general, the incidence and mortality rates of iCCA were reported rising worldwide in the past decade, whereas the incidences of pCCA seem to be stable or decreasing[10]. Besides the burden of HCV infection has been linked with the incidence of iCCA, infection with liver flukes and HBV-related liver diseases have shown a stronger association for iCCA in East Asia[10,11]. Primary sclerosing cholangitis (PSC) is considered as a predominant risk factor for CCA, especially for pCCA[10,12]. The pronounced geographical and etiology heterogeneity of iCCA and pCCA indicated the potential diverse cancer-initiating cell in these two subtypes of CCA[13]. In detail, iCCA is considered to originate two different cell of origin (mucin-secreting cholangiocyte or hepatic progenitor cell), due to the significant intertumor heterogeneity[14]. Unlike iCCA, pCCA only originated from mucin-secreting cholangiocyte[13,15].

To date, previous studies of the genomic alterations in a variety of bile duct cancer, especially for iCCA, have demonstrated some known commonalities in the mutation of CCA. These studies identified a number of driver genes, but most of them represented the genomic features of iCCA, such as TP53, KRAS, SMAD4, and ARID1A[16–20]. Notably, the latest integrative genomic analysis of Caucasian extrahepatic CCA (eCCA) (including the pCCA and dCCA) revealed distinct molecular characterization of eCCA in West[21]. However, the epidemiological profile of CCA and its subtypes showed enormous geographical variation, indicating underlying genomic heterogeneity across different regions[5]. The prevalence of CCA is highest in East Asia and the subtype distribution is similar. Thus, integration analysis in sequencing data from Asian countries may find significant genomic features ignored by previous studies.

Here, we perform whole-exome sequencing on 43 pCCA and 24 iCCA, and analyze an additional 44 pCCA and 237 iCCA to investigate the genomic landscapes of pCCA and iCCA, respectively. Our study identifies significant mutated CCA driver genes and emphasizes the similar and different genomic characteristics between pCCA and iCCA. Notably, we highlight the effect of underlying post-transcriptional modification in CCA occurrence and development, and demonstrate that METTL14 and its loss-of-function mutation R298H play an important role in CCA process through N6-methyladenosine (m6A) mRNA methylation on the potential driver gene MACF1. Our results may be valuable for better understanding of post-transcriptional modification effects CCA development, and the similarities and differences between pCCA and iCCA.

## Results

**Mutational spectrum of pCCA and iCCA.** We conducted integration analysis of exome-sequencing data from 87 pCCA cases (including 43 Nanjing Medical University (NMU) cases and 44 International Cancer Genome Consortium (ICGC) cases) and 261 iCCA cases (including 24 NMU cases, 135 ICGC cases, and 102 cases from Zou et al.'s study[17]). The general information and clinical features of patients from the NMU study are listed in Supplementary Table 1. The mean sequencing depth of the NMU study was 94.4× in tumor tissues and 91.8× in normal tissues dissected adjacent to the tumor. Detailed sequencing coverage and depth information are listed in Supplementary Table 2. We detected a total of 91,995 somatic mutations (Supplementary Table 2) in all the cases, and the median number per case was 120.5. The median tumor mutation burden (TMB) was 2.0 megabase per case (/Mb) and the burden was comparable between pCCA (2.0/Mb) and iCCA (2.0/Mb) (Wilcoxon rank-sum test $P = 0.48$, Supplementary Fig. 1A). Among these mutations, we defined 28,254 nonsynonymous mutations, for which the median was 33 single nucleotide variants (SNVs) and 3 small insertion and deletions (INDELs) per patient. Interestingly, the nonsynonymous mutation burden (median 0.61/Mb) of iCCA was significantly higher than that (median 0.47/Mb) of pCCA (Wilcoxon rank-sum test $P = 9.6 \times 10^{-4}$, Fig. 1A). Similar to the non-synonymous mutation pattern, iCCA had a higher copy number alteration burden (CNAB) than pCCA (iCCA median = 25.0%, pCCA median = 8.4%, Wilcoxon rank-sum test $P = 2.4 \times 10^{-4}$), no matter the amplification or deletion (Fig. 1C). The CNAB based on purity-adjusted copy number in iCCA was also significantly higher than it in pCCA (All burden: Wilcoxon rank-sum test $P = 8.0 \times 10^{-4}$; Amplification burden: Wilcoxon rank-sum test $P = 1.9 \times 10^{-6}$; Deletion burden: Wilcoxon rank-sum test $P = 4.3 \times 10^{-2}$; Supplementary Table 3).

Next, we investigated the genomic mutational signatures, which provide clues for the carcinogenesis of CCA. Firstly, we classified all the mutations into six classic subtypes. Consistent with previous studies[17,18], C > T/G > A substitutions accounted for the predominant type of all SNVs in all of the patients (Supplementary Fig. 1B). The proportion of the mutation subtypes was similar between pCCA and iCCA (Wilcoxon rank-sum test $P = 0.46$, 0.09, 0.67, 0.62, 0.51, and 0.29 respectively for C > A, C > G, C > T, T > A, T > C, and T > G, Supplementary Fig. 1B). We further analyzed mutational signatures using R package SignatureEstimation, which accurately reconstructs the mutation profiles of all pCCA and iCCA patients based on a predefined mutational spectrum of 30 COSMIC signatures[22]. COSMIC signature 1 (Age, 44.71%), signatures associated with mismatch repair deficiency (MMR, Signatures 6, 15, and 26, 31.99%), signatures associated with activated APOBECs (Signatures 2 and 13, 6.48%), and signature 8 (6.41%) were dominant (proportion > 5%) in pCCA patients (Fig. 1B). For iCCA patients, COSMIC signature 1 (27.19%) and signature MMR (26.74%) were also identified as dominant, but the proportion was slightly lower than in pCCA patients. In addition, signature 4 related to smoking behavior (10.13%), signature 22 (9.96%) related to aristolochic acid, and signatures specific to liver cancer (Liver, Signatures 12, 16, and 24, 8.7%) were predominant in iCCA. Interestingly, signature Liver were more common in HBV-related iCCA (HBV: 26.0%, non-HBV: 3.8%, Supplementary Table 4). Aristolochic acid-related mutations were mainly found in iCCA patients from Zou et al.'s study, but this signature was also observed in pCCA patients (Supplementary Table 5). In addition, signature 3, related to homologous recombination deficiency (4.26%), and signature 10, related to altered POLE activity (2.44%) were only identified in iCCA (Fig. 1B), which suggested that multiple types of DNA damage occurred during the carcinogenesis progress of CCA.

The results mentioned above revealed the nature of genomic instability in CCA patients. We further investigated the association between TMB/CNAB and the overall survival of patients. We

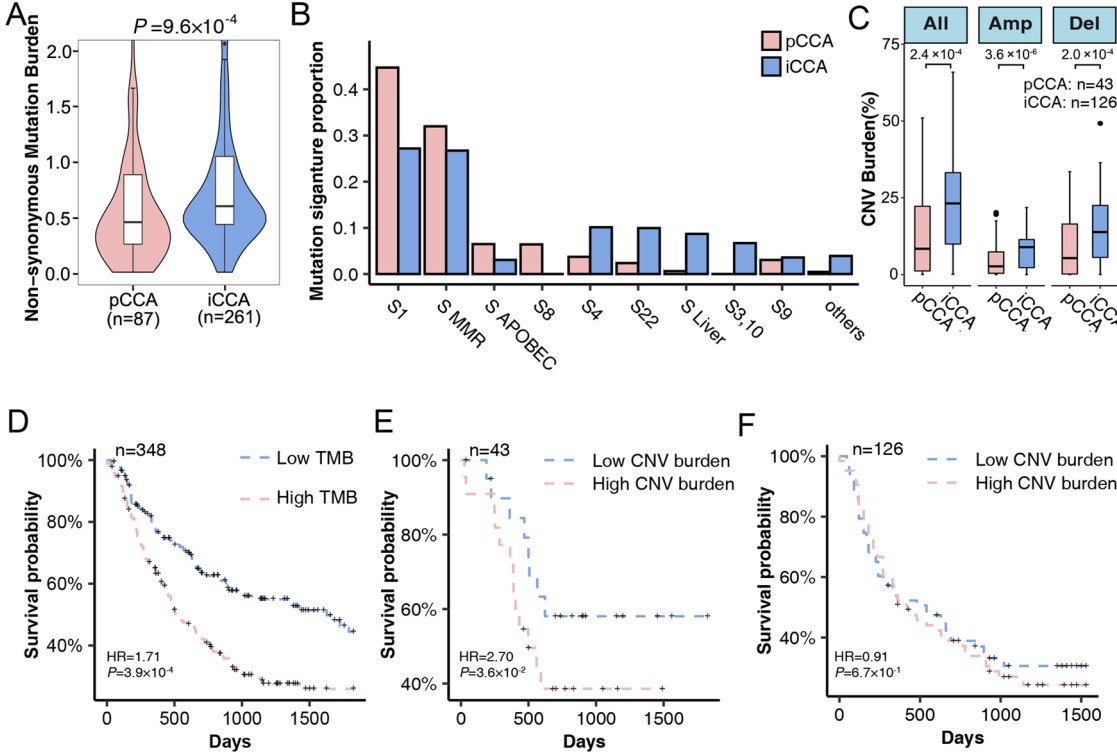

**Fig. 1 Mutational spectrum of patients with CCA. A** The tumor mutation burden (TMB) of nonsynonymous mutations (substitutions and indels) between pCCAs (presented in red) and iCCAs (presented in blue). $P$ value was obtained from Wilcoxon rank-sum test. **B** Summary composition of COSMIC 30 mutation signatures in pCCAs (red) and iCCAs (blue). COSMIC Signature MMR included signatures 6, 15, and 26; Signature liver included signatures 12, 16, and 24; Signature APOBEC included signatures 2 and 13. **C** Difference of copy number alterations burden (CNAB) no matter amplification or deletion between pCCAs (red) and iCCAs (blue). Wilcoxon rank-sum test was performed to determine the $P$ value. Box plots depicted the median, quartiles and range. The whiskers in box plots extended to the most extreme data point which is no more than 1.5 times IQR. Outliers were identified using upper/lower quartile ±1.5 times IQR. **D** Cox proportional hazard model adjusted for age, gender, and stage was conducted to obtain HR and $P$ value ($n = 348$). **E**, **F** Prognostic effect of CNAB in all samples. **E** CNAB in 43 pCCAs. **F** CNAB in 126 iCCAs. *$P < 0.05$; **$P < 0.01$; ***$P < 0.001$; TMB Tumor mutation burden, CNV Copy number variation, Amp Amplification, Del Deletion.

found that the patients carrying higher loads of mutations showed worse outcomes after adjusting for age, sex, and stage at diagnosis (Hazard ratio (HR) = 1.71, $P = 3.9 \times 10^{-4}$, Fig. 1D). The association was consistent in pCCA and iCCA patients (Supplementary Fig. 1C). When examining CNAB, we noticed that the survival of pCCA patients with higher CNAB was significantly worse than for patients with lower CNAB (HR = 2.70, $P = 3.6 \times 10^{-2}$, Fig. 1E); However, there was no significant difference between iCCA patients with distinct CNAB (HR = 0.91, $P = 6.7 \times 10^{-1}$, Fig. 1F). Consistent findings were obtained from the prognostic effect of CNAB based on purity-adjusted CNAB (pCCA: HR = 2.89, $P = 3.6 \times 10^{-2}$; iCCA: HR = 0.84, $P = 4.5 \times 10^{-1}$; Supplementary Table 6).

**Crucial potential driver genes in CCA.** We identified 36 significantly mutated genes (SMGs, Fig. 2), also described as mut-drivers, in all patients by MutSig2CV and IntOGen, including 24 reported SMGs and 12 SMGs not highlighted in previous CCA publications (Supplementary Table 7). Among these SMGs, MACF1 (5.17%, 18/348) and AXIN1 (0.86%, 3/348, all of the three patients were with iCCA), which is commonly considered as one of the potential drivers in hepatocellular carcinoma (HCC), are two interactive components of the β-catenin/Wnt signaling pathway (Fig. 2). Although the mutation rate of AXIN1 was low, we noticed that there were multiple non-synonymous mutations in two iCCA patients (Fig. 2). The mutations of PIK3R1, a component of the PI3K pathway, was displayed in 2.59% (9/348)

of patients. Post-transcriptional modification genes RBM10 (3.16%, 11/348) and METTL14 (0.86%, 3/348) act as well-known alternative splicing regulators and m6A writers. Importantly, we identified recurrent mutations in METTL14, and all mutations in this gene affected the same amino acid (Supplementary Fig. 2B), which were also predicted as potentially deleterious variants by several bioinformatic pathogenicity prediction tools (Supplementary Table 7). We observed the same mutation rate (1.44%, 5/348) of chromatin modifiers SMARCA4 and WHSC1. Interestingly, mutations in these genes were prone to co-occur in the same patient (Fisher's exact test OR = 660, $P = 2.29 \times 10^{-6}$, Fig. 2). In addition, classic cancer driver genes identified in other cancers were also identified as SMGs in this study, including ATM (3.45%, 12/348), BRCA2 (2.01%, 7/348), and MLLT4 (2.01%, 7/348). Although BRCA2, EPHA2, and ATM were mentioned in previously published work due to their same position recurrent inactivating mutations, our integrated analysis with more sample size identified them as SMGs.

We further defined the genes with a distinct mutation rate between pCCA and iCCA. TP53, ARID1A, PBRM1, MACF1, EPHA2, ARID2, IDH1, PTEN, RB1, BRAF, NRAS, SLC8A1, AXIN, and MLLT4 were iCCA-enriched genes, while RBM10, TGFBR2, PIK3R1, ELF3, NACC1, and METTL14, were pCCA-enriched genes. The mutation rates of TP53 and IDH1 were significantly higher in iCCA (Fisher's exact test $OR_{TP53} = 1.3$, $P_{TP53} = 8.8 \times 10^{-3}$; $OR_{IDH1} = +\infty$, $P_{IDH1} = 4.4 \times 10^{-2}$, Fig. 2). Importantly, IDH1 only mutated in iCCA, but not in pCCA. RBM10, however, carried significantly more mutations in

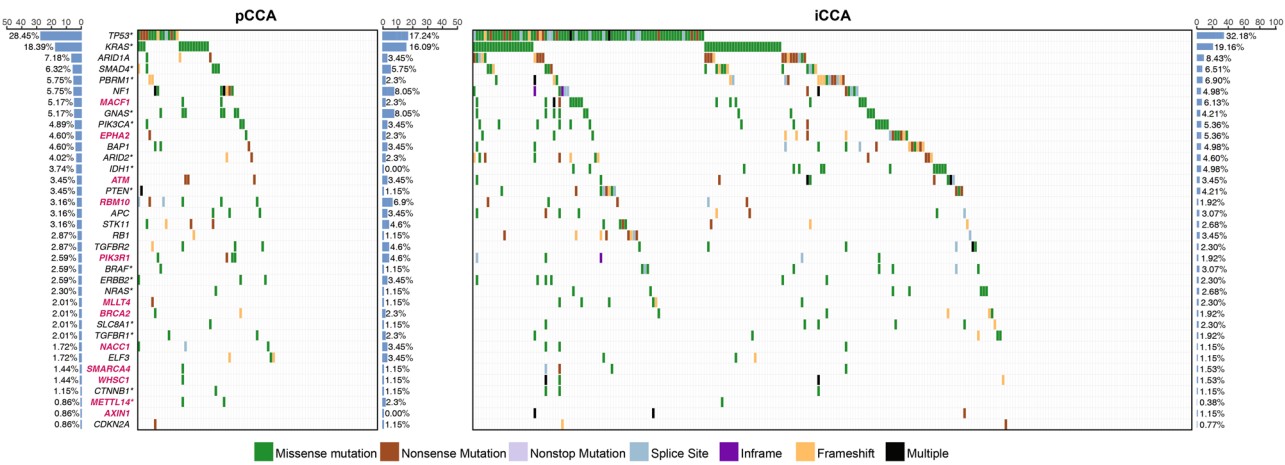

**Fig. 2 Mutation landscapes of curated significantly mutated genes (SMGs) and recurrent mutation genes in CCA.** Each column represents one CCA patient and each row represents a gene. The gene symbols in red are our newly identified CCA-related genes, and genes in black have been reported in CCA. The bar chart at the left panel indicates the overall mutational rate of curated genes in all CCAs. The right panel is the mutation spectrum and mutation rate for pCCAs and iCCAs, respectively. The asterisks following gene symbol indicate there are recurrent mutations in the relevant genes.

pCCA than in iCCA (Fisher's exact test $OR_{RBM10} = 3.8$, $P_{RBM10} = 3.2 \times 10^{-2}$).

We further identified 22 regions by focal copy number alteration (CNAs) in CCAs (Fig. 3A, Supplementary Table 8), including 16 regions overlapping with previously reported peaks, and six significantly altered regions not reported before (amplification at 7q31.2, 22q11.21, and deletion at 1p36.13, 2p24.1, 7q35, and 12q24.33, Fig. 3A). The cancer-related genes located at these peaks were considered to be CNA-driver genes. The gain of a 0.8-Mb region at 7q31.2 occurred in 45 samples (26.6%), which contains the classic MET oncogene in the RTK/RAS pathway. However, the amplification of 7q31.2 was found rare previously in Caucasian population. The loss at 1p36.13 involved tumor suppressor gene SHDB and affected 76 samples (45.0%). DNA Polymerase Epsilon (POLE) at 12q24.33, where was also considered lost in HCC, lost one copy in 40 patients (23.7%). In addition, important DNA methylation factor DNMT3A (2p24.1) and chromatin methylation factor EZH2 (7q35) also show loss in 20 (11.8%) and 19 (11.24%) patients (including iCCA and pCCA), respectively. Whereas these two regions were also only considered deleted in iCCA based on a previous single nucleotide polymorphism array analysis. We only observed heterogeneous loss in CCA patients (Fig. 3A). Among all of the regions, the frequencies of amplification/gain at 8q24.21 (MYC), 1q21.3 (S100A7), 5p15.33 (SDHHA), and 7q31.2 (MET) were significantly higher in iCCA (Supplementary Table 8). The frequency of high-level amplification was significantly higher in pCCA than in iCCA, although the frequency of amplification/gain was comparable across the subtypes (Fig. 3A). In addition, we found that copy number loss and nonsynonymous mutations of classic tumor suppressors (TP53, CDKN2A, SMAD4, PTEN, and ATM) were prone to co-occur in the same patients (Fisher's exact test OR = 8.4, $P = 8.2 \times 10^{-6}$, Fig. 3B).

Next, we performed pathway enrichment analysis on all potential driver genes identified in this study, which included SMGs (mut-drivers) and cancer genes in the frequently altered focal CNA regions (CNA-drivers) described above, and found that the potential driver genes of CCA were significantly enriched in the RTK-RAS, Wnt, PI3K, Cell Cycle, TP53, TGF-beta, and HIPPO pathways (Fig. 3C). Including the identified MET amplification, 34.3% of CCA patients harbored mutations and CNAs in oncogenes from the RTK-RAS pathway, and these alterations occurred in a mutually exclusive manner, as reported in other cancers (Supplementary Fig. 2A).

**Functional recurrent mutations in METTL14.** Among the potential driver genes mentioned above, we identified a potential driver gene, METTL14, the main factor involved in aberrant m6A modification of various cancers, with recurrent and deleterious mutations[23,24] (Supplementary Fig. 2B, Supplementary Table 9). All three mutations affected the same 298 amino acid residue (METTL14 p.R298H and p.R298C) and two of them were in the same genomic position (two patients from NMU and ICGC respectively) (Supplementary Fig. 2B). We further conducted sanger sequencing in an independent pCCA cohort with extra 40 subjects from NMU cohort and identified an additional p.R298H carrier (Supplementary Fig. 2C, D). In the COSMIC database, we also found that the same p.R298H also occurred in three pancreatic ductal carcinomas (Supplementary Fig. 2B) and all the three patients were East Asian.

Crystal structures of the METTL3-METTL14 complex have revealed that p.R298 lies close to the putative RNA-binding groove of the complex which may have a complex role to affect methylation activity[25]. The recurrent and deletions mutant p.R298H suggested us its possibly positive selection and the need of METTL14's normal action in antitumor progress. However, the relevance of this hotspot mutation and m6A mRNA methylation to the CCA has not yet been established. Dysfunction of METTL14, the key catalytic protein forming the core m6A methyltransferase complex, has shown fundamental biological effects in cancer initiation and progression[24,26]. Thus, we hypothesized that CCA could be associated with METTL14 that regulated m6A mRNA methylation. Hence, we first examined METTL14 expression in 69 CCA pairs and matched adjacent normal tissues. The results of qRT-PCR and western blot revealed that the expression of METTL14 was significantly downregulated in tumors ($P < 0.01$; Fig. 4A, B and Supplementary Fig. 7A). The immunohistochemistry (IHC) staining on tissue microarray also confirmed that METTL14 staining was decreased in CCA at the protein level (Supplementary Fig. 3A) and its downregulation displayed a significant association with poor cancer-specific survival in CCA (Log-Rank $P = 0.045$; Supplementary Fig. 3B). Consistently, we found that the m6A level of total RNA was significantly decreased in CCA tissues (Fig. 4C). Interestingly, we also observed the m6A modification level was significantly decreased in METTL14 low expression CCA group compared with METTL14 high expression CCA group (Supplementary Fig. 3C). These results suggested that METTL14 and it mediated m6A modification were frequently downregulated or disturbed in CCA.

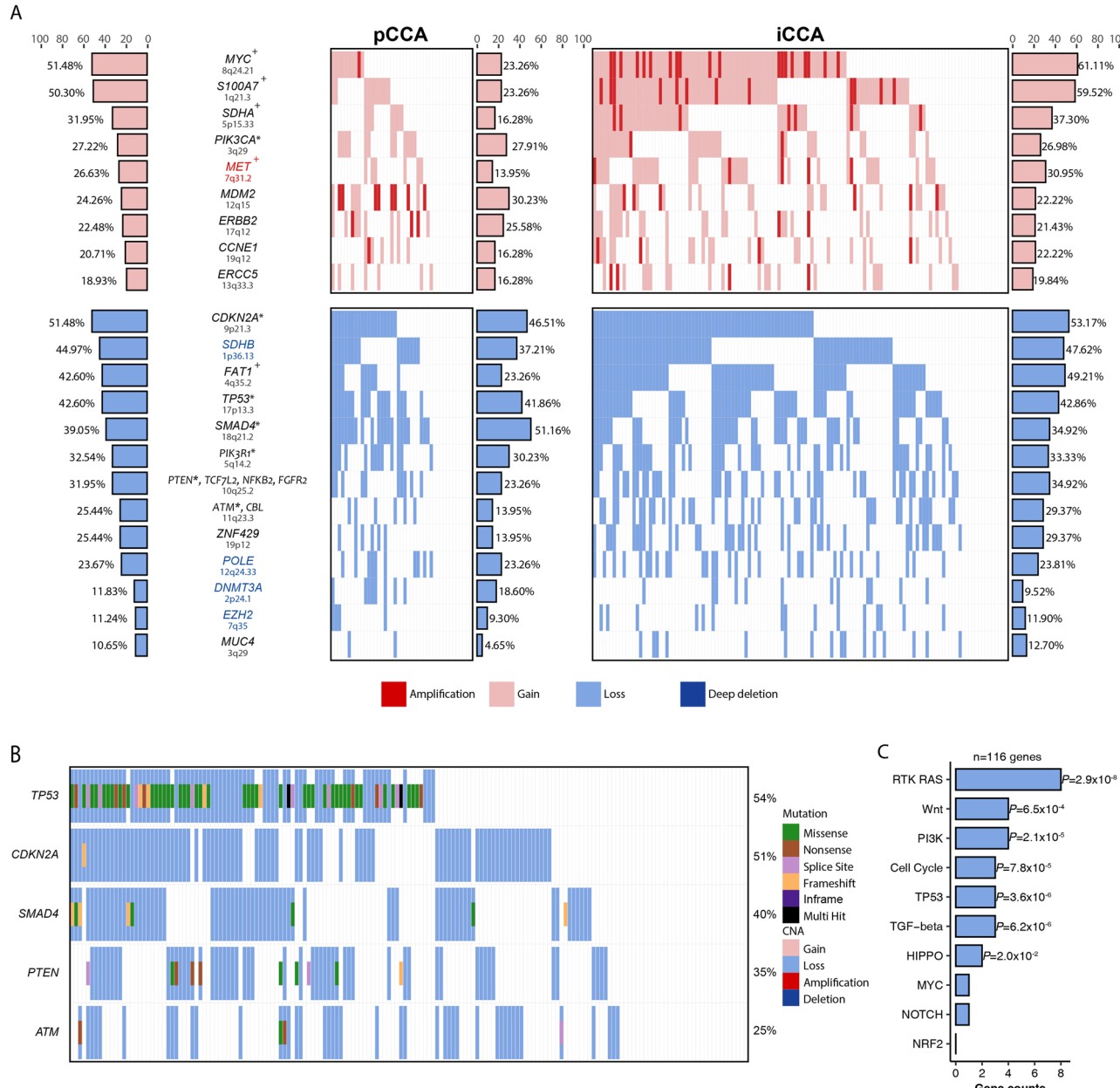

**Fig. 3 Frequently altered focal CNA regions and characteristics of mutation-driver and CNA-driver genes. A** Shows 22 curated frequently-altered focal CNA regions involving cancer-related regions. Each column represents one of the CCA patients and each row represents a focal region. The bar chart at the left panel indicates the overall CNA rate in all CCAs. The right panel is the CNA spectrum and rate for pCCAs and iCCAs, respectively. The top panel represents amplification and the bottom represents deletion regions. Region names in red and blue indicate the novel alterations the plus sign following the gene symbol indicates a significant alteration rate difference between pCCAs and iCCAs. Asterisk means the genes in focal region were also identified in SMGs. **B** The co-occurrence of copy number losses and mutations of relevant tumor suppressor genes. **C** Pathway enrichment results of mutation-driver and CNA- driver genes in TCGA pancancer ten pathways, *P* value of hypergeometric test. *P < 0.05; **P < 0.01, ***P < 0.001.

To determine the "driver" role of METTL14 and R298H mutations during CCA development, we first verified that there were no R298H mutations of METTL14 in CCA cell lines (RBE and HCCC9810) using sanger sequencing (Supplementary Fig. 3G), and then confirmed the transfection efficiency of lentiviral constructs expressing METTL14wt and METTL14R298H in RBE and HCCC9810 cell lines (Fig. 4D, Supplementary Figs. 3D, E, and 7C). In contrast to the cells stably overexpressing METTL14wt, cells overexpressing METTL14R298H showed significantly decreased overall m6A modification ability (Fig. 4E and Supplementary Fig. 3F). Subsequently, while overexpression of wild-type METTL14 affected cell proliferation and apoptosis,

overexpression of the mutation had no noticeable effect on proliferation and apoptosis (Fig. 4F–H, Supplementary Figs. 3H, I, L, and 4A). In addition, overexpression of METTL14wt resulted in a decrease in cell migration and invasion, and METTL14R298H remarkably reversed the gene's ability to inhibit migration and invasion in CCA cells (Fig. 4I and Supplementary Fig. 3J, K). Taken together, these results provided us evidence for loss of function caused by METTL14R298H mutation, suggesting that METTL14R298H mutation could reduce the tumor-suppressive effect of METTL14wt in CCA.

To further verify the functional roles of METTL14wt and METTL14R298H in vivo, we observed that METTL14wt effectively

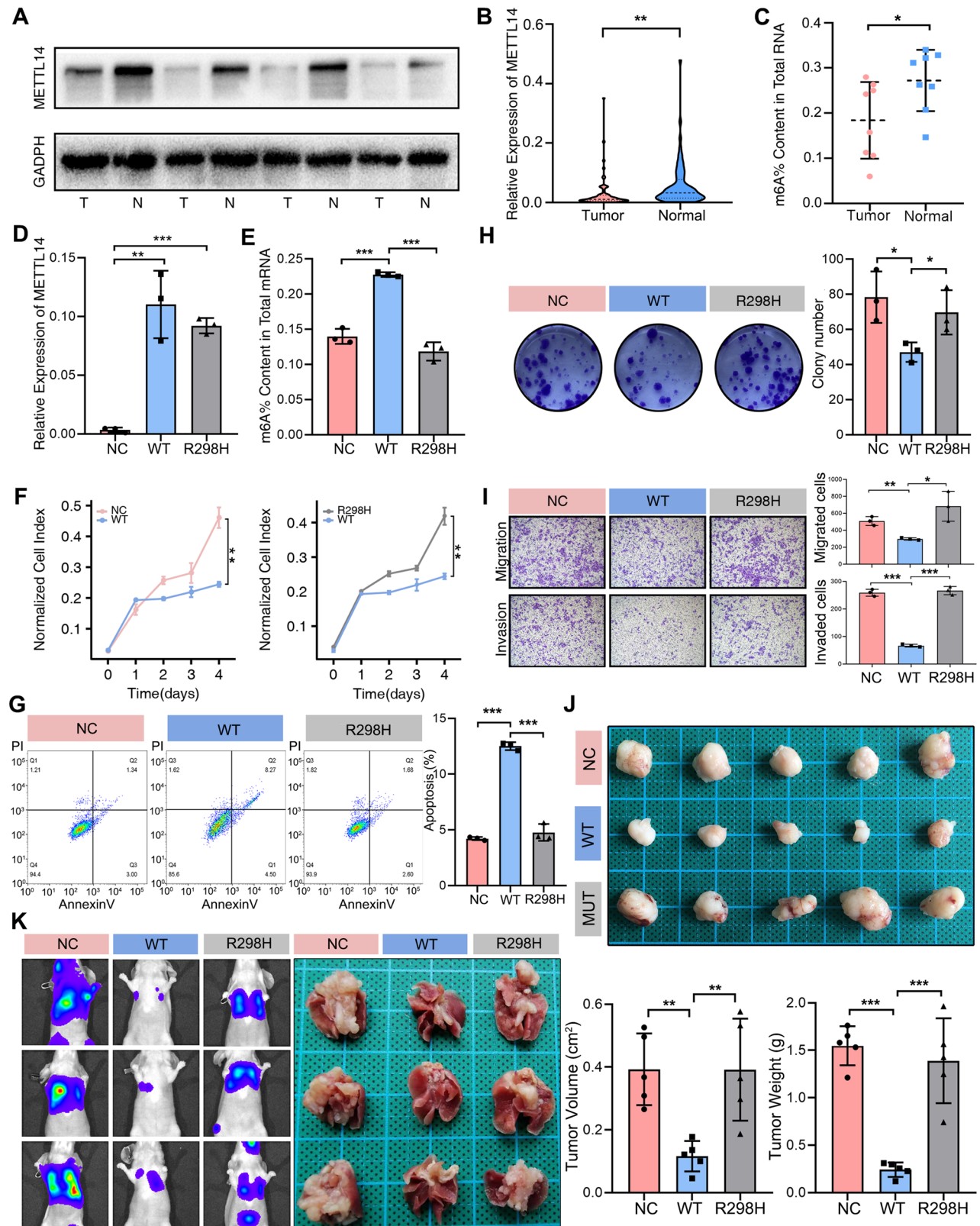

suppressed tumor growth as reflected by the significant reduction of tumor size and weight in nude mice, and in a lung metastasis model, METTL14$^{wt}$ could also restrain pulmonary metastasis significantly. Consistent with results from in vitro experiments, the inhibition of METTL14$^{R298H}$ on tumor proliferation and lung metastasis was apparently attenuated in vivo when compared to the METTL14$^{wt}$ group (Fig. 4J, K). Therefore, these data implied

that loss of function caused by METTL14$^{R298H}$ mutation was sufficient to repress the tumor suppressor function of METTL14$^{wt}$ in CCA.

**Potential driver gene MACF1 served as the target of METTL14.** We then conducted RNA-Seq and MeRIP-Seq assays in negative

**Fig. 4 METTL14^(R298H) mutation reduced the METTL14^(wt) tumor-suppressive effect in CCA. A** The protein level of METTL14 was validated in CCA tissues. **B** Downregulated METTL14 mRNA expression was detected in 69 pairs of iCCA and pCCA tumor tissues as well as the adjacent normal tissues by qRT-PCR ($p = 0.0064$). **C** The level of m^6A modification in tumor and adjacent normal tissues ($n = 8$). **D** The transfection efficiency of lentiviral constructs expressing METTL14^(wt) and METTL14^(R298H) in RBE cell line ($n = 3$). **E** METTL14^(R298H) reduced METTL14^(wt)-mediated m^6A modification detected by m^6A colorimetric quantification in RBE cell line ($n = 3$). **F** Proliferation curve of RBE cells with METTL14^(R298H), METTL14^(wt) or negative control ($n = 3$). **G** Apoptotic assay of RBE cells with METTL14^(R298H), METTL14^(wt), or negative control were determined by a PI and annexin V double-staining assay and analysis by flow cytometry ($n = 3$); p(NC vs WT) < 0.001, p(WT vs R298H) < 0.001. **H** Colony formation assay of RBE cells with METTL14^(R298H), METTL14^(wt), or negative control. The numbers of colonies were counted and presented in the histogram ($n = 3$). **I** Representative images (left) and quantification (right) of transwell migration (p(NC vs WT) = 0.0024; p(WT vs R298H) = 0.0194) and invasion (p(NC vs WT) < 0.001, p(WT vs R298H) < 0.001) assays in RBE cells with METTL14^(wt), METTL14^(R298H) or negative control ($n = 3$). **J** Top panel is a photograph of excised tumor tissues from mice. Bottom panel is the average weight (p(NC vs WT) = 0.0011; p(WT vs R298H) = 0.0066) and volume (p(NC vs WT) < 0.001; p(WT vs R298H) = 0.0005) of harvested tumors in the NC, METTL14^(wt) and METTL14^(R298H) groups ($n = 5$). The data are presented as the mean ± standard deviation. **K** METTL14^(R298H) reversed the tumor-suppressive effect of METTL14^(wt) in the lung metastasis experiment in vivo. Left panel is representative images taken at six weeks in NC, METTL14^(wt) and METTL14^(R298H) groups (n = 3). Data are shown as mean ± SEM. The P values were calculated using unpaired two-sided Student's t test with no correction for multiple comparison. *$P < 0.05$, **$P < 0.01$, ***$P < 0.001$; T, Tumor tissue, N, normal tissue, R298H, METTL14^(R298H); WT, METTL14^(wt); NC, negative control.

control, METTL14^(wt), and METTL14^(R298H) cells. A total of 2,601 peaks involving classic transcripts of 1,586 genes were robustly identified by exomePeak2 in all cells. The most common m^6A motif, GGAC, was significantly enriched in the m^6A peaks identified (Supplementary Fig. 4B), and the m^6A peaks were especially enriched in the vicinity of the stop codon. Next, we performed differential m^6A-methylation analysis between control and METTL14^(wt) cells, as well as between METTL14^(wt) and METTL14^(R298H) cells. Because of the writer role of METTL14 during the m^6A methylation modification, we included only m^6A peaks (712) with increased abundance in METTL14^(wt) cells as compared to control cells, and peaks (990) with decreased abundance in METTL14^(R298H) as compared to METTL14^(wt) cells. A total of 237 peaks were shared by both analyses, including four peaks on the two potential driver genes mentioned above (MACF1, MET) (Fig. 5A, B, Supplementary Fig. 4C). MACF1 was also the potential driver gene identified in this study (Supplementary Table 7). We then verified whether m^6A-modified MACF1 was susceptible to decay. The lifetime of MACF1 was prolonged in METTL14^(R298H) cells and shortened in METTL14^(wt) cells after actinomycin D treatment (Fig. 5C, and Supplementary Fig. 4D). We further immunoprecipitated m^6A from the RNAs of METTL14^(wt) and METTL14^(R298H) cells and found that METTL14^(R298H) significantly decreased the amount of MACF1 modified by m^6A compared to METTL14^(wt) (Fig. 5D, and Supplementary Fig. 4E). According to the previous study, p.R298 of METTL14 belongs to the groove between METTL3 and METTL14 which might be responsible for RNA binding, hence we supposed that METTL14^(R298H) may regulate the enrichment of MACF1 mRNA[25,27]. And we found that METTL14-specific antibody obversely enriched MACF1 mRNA in METTL14^(wt) compared to the METTL14^(R298H) group using RNA immunoprecipitation (Fig. 5E, and Supplementary Fig. 4F). We then applied immunofluorescence assays and western blot to further confirm that METTL14^(wt) mediated MACF1 degradation, and METTL14^(R298H) showed increase expression of MACF1 compared to METTL14^(wt) (Fig. 5F, G, Supplementary Figs. 4H, I, and 7B). We also found that METTL14^(wt) cells decreased the expression of MACF1, and no noticeable effect on MACF1 expression was observed in METTL14^(R298H) cells using qRT-PCR (Supplementary Fig. 4J).

When we successfully transfected CCA cells with siRNA pools targeting MACF1 (Supplementary Figs. 4K, L, and 7D), we found that knockdown of MACF1 significantly increased cell apoptosis and reduced CCA cell proliferation and metastasis in functional assays (Fig. 5H–K, and Supplementary Fig. 5A–F). Recent studies reported that MACF1 was involved in tumor metastasis and cytoskeleton, and that it played crucial roles in the nucleus

translocation of β-catenin[28,29]. In our findings, we first validated MACF1 upregulation in CCA using qRT-PCR(Supplementary Fig. 4G), and noticed that lower METTL14 mRNA level exhibited stronger MACF1 expression (Fig. 5L). Given the essential role of MACF1 in regulating the nucleus translocation of β-catenin, immunofluorescence assays and western blotting analysis showed that the increase of nuclear β-catenin was correlated with the expression of METTL14^(R298H) rather than METTL14^(wt) (Fig. 5M, N, Supplementary Figs. 5G, H, and 7E). To ascertain the role of MACF1 in METTL14-mediated nucleus translocation of β-catenin, we transfected MACF1 siRNA in METTL14^(R298H)-overexpressing cells, and observed that the nucleus translocation of β-catenin was decreased compared to only METTL14^(R298H)-overexpressing cells (Supplementary Fig. 5I).

Additionally, METTL14^(R298H) resulted in reversing the expression level of E-cadherin, N-cadherin, PCNA, and CyclinD1, which were the downstream targets of β-catenin, relevant to METTL14^(wt) (Fig. 5N, Supplementary Figs. 5G, and 7B, E). These results implied that METTL14-mediated m^6A modification repressed the MACF1/β-catenin pathway in CCA, while METTL14^(R298H) mutation disrupted this mechanism.

## Discussion

In this study, we included genomic data from 348 CCA samples (including 87 from pCCA and 261 from iCCA) to investigate the genomic landscapes of both pCCA and iCCA. We found the shared and distinct features between these two anatomical subtypes. Currently, there is no effective biomarker that can accurately predict the prognosis of CCA patients[30,31]. Our study found that TMB has a good predictive effect on the prognosis of East-Asian CCA patients. It is worth noting that pCCA patients with higher CNAB showed worse survival outcomes compared to those with lower CNAB. These findings suggest that we can explore prognostic markers of patients with CCA from a more macroscopic genetic perspective.

In addition to several significant driver genes in classic pathways, we identified driver genes participating in post-transcriptional modification (i.e., METTL14 and RBM10), which carried more mutations in pCCA patients. With in vitro experiments, we confirmed the role of METTL14, as well as the role of this mutation in CCA development. We further determined that m^6A modification level and expression of potential driver gene MACF1 could be regulated by METTL14, which can influence the proliferation and metastasis ability of CCA cells. Notably, although we provide evidence to demonstrated that METTL14-mediated m^6A modification repressed the MACF1/β-catenin pathway in CCA, while METTL14^(R298H) mutation

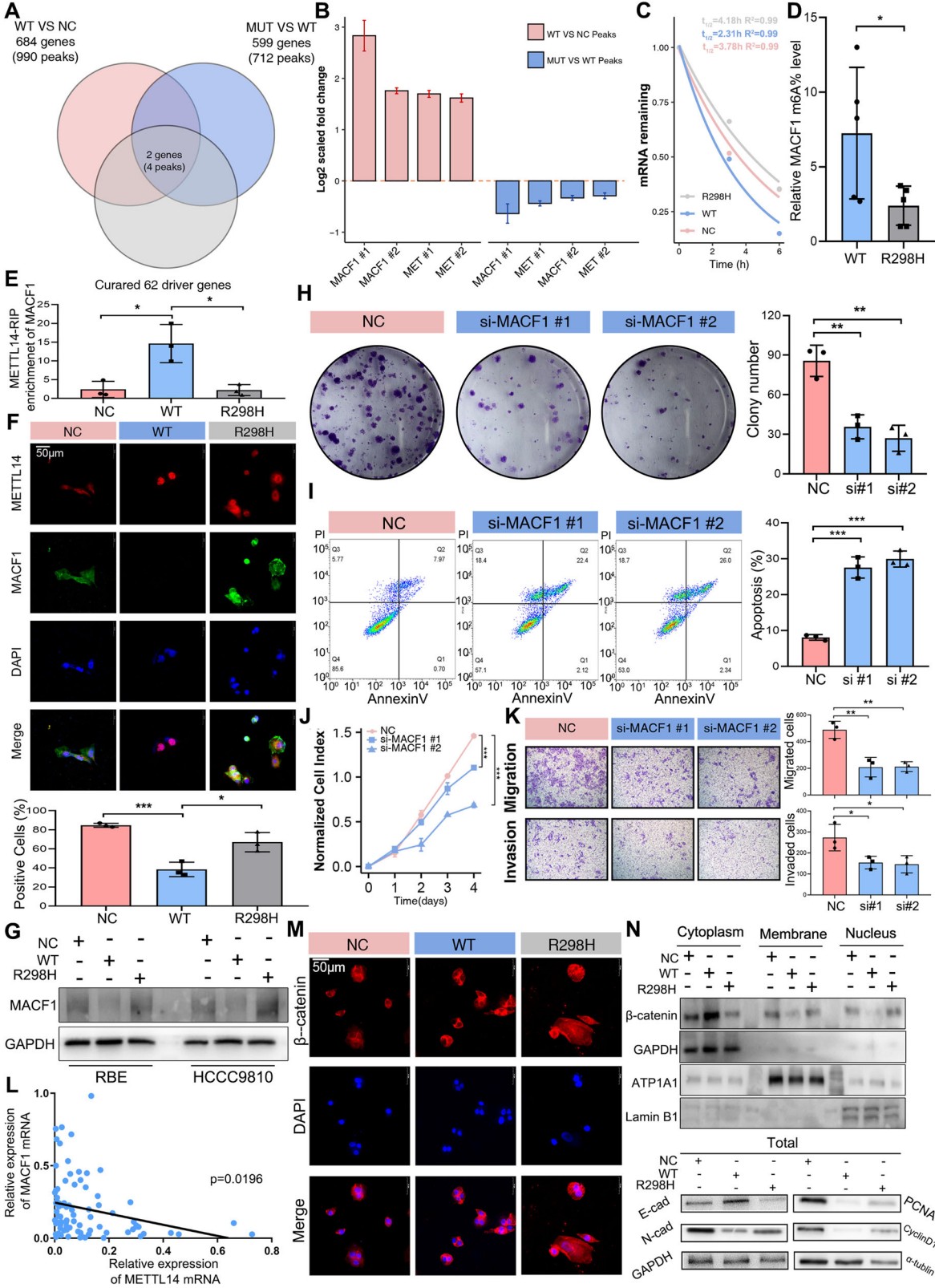

disrupted this mechanism, m6A-related key catalytic protein would be more complicated on the WNT pathway and the underlying mechanism is not fully characterized. Previously, ALKBH5 reducing m6A levels of WIF-1 and hindering activation of Wnt signaling[32]. YTHDF bound METTL3-mediated m6A of APC mRNA and upregulated Wnt/β-catenin pathway[33]. Given the importance of Wnt signaling in regulating cancer

tumorigenesis, METTL14-MACF1-WNT would be more complicated, which deserves further investigation.

The previous study showed that there was lack of evidence that anatomical site determines molecular subtypes according to integration analysis of transcriptomic and methylation data[34]. Our study identified four potential driver genes that were related to chromatin and methylation modification (i.e., mut-drivers:

**Fig. 5 M⁶A modification mediated expression of MACF1 in β-catenin nucleocytoplasmic transport. A** Overlap of significantly different m$^6$A modification peaks identified from METTL14$^{R298H}$ compared with METTL14$^{wt}$, METTL14$^{wt}$ compared with NC and mutation-driver genes. **B** The log2 scaled fold change of different m$^6$A modification peaks in WT versus NC and R298H versus WT. **C** RNA lifetime for MACF1 in RBE cells transfected with METTL14$^{R298H}$, METTL14$^{wt}$, or NC. **D** The m$^6$A modification level of MACF1 in RBE cells was validated in MeRIP ($n = 5$). **E** Immunoprecipitation of METTL14-related RNA in NC, METTL14$^{wt}$ and METTL14$^{R298H}$ was conducted in RBE cells followed by RT-qPCR to detect the amount of MACF1 mRNA binding to METTL14 ($n = 3$). **F** Representative images of MACF1 immunofluorescence in METTL14$^{wt}$ and METTL14$^{R298H}$ ($n = 3$). **G** Expression of MACF1 following NC, METTL14$^{wt}$ and METTL14$^{R298H}$ was evaluated by western blotting. All data are representative of at least two independent experiments with similar results. **H** Colony formation assay of RBE cells with NC and si-MACF1 ($n = 3$). **I** Apoptotic assay of RBE cells with NC and si-MACF1 were determined ($n = 3$; p(NC vs si#1) = 0.0004; p(NC vs si#2) < 0.0001). **J** Proliferation curve of RBE cells with MACF1 and si-MACF1 ($n = 3$; p(NC vs si#1) < 0.0001; p(NC vs si#2) = 0.0001). **K** Transwell migration and invasion assays in RBE cells with MACF1 and si-MACAF1 ($n = 3$; Migration:p(NC vs si#1) = 0.0076; p(NC vs si#2) = 0.003; Invasion: p(NC vs si#1) = 0.0417; p(NC vs si#2) = 0.0430)). **L** QRT-PCR results revealed that expression of METTL14 and MACF1 was negatively interrelated in CCA tissues ($n = 82$). **M** Representative images of β-catenin immunofluorescence showed nuclear β-catenin tended to be increased by expression of METTL14$^{R298H}$ compared to METTL14$^{wt}$ in RBE cells. All data are representative of at least two independent experiments with similar results. **N** Western blot analysis was performed to analyze β-catenin in the cytoplasmic, membrane and nuclear extracts, and expression of E-cadherin, N-cadherin, PCNA, and CyclinD1 in total RBE cells extracts. All data are representative of at least two independent experiments with similar results. The *P* values were calculated using unpaired two-sided Student's *t* test with no correction for multiple comparison. Data are shown as mean ± SEM. *P < 0.05, **P < 0.01, ***P < 0.001.

SMARCA4 and WHSC1; CNA-drivers: DNMT3A and EZH2), further supporting the finding that driver genes cause disruption at in the transcriptional level. Consistent with the previous study's findings, we found comparable frequency between these alterations of potential driver genes. However, our analysis identified two genes as drivers (METTL14 and RBM10) and found that they are related to post-transcription modification. Mutations in these two genes were more common in pCCA patients. This evidence suggested that the investigation of the difference between pCCA and iCCA should be extended to the post-transcriptional area. METTL14 engages in m$^6$A modification[35,36], which modulates alternative splicing, export, stability, and translation of mRNA[37]. Interestingly, we found that two potential driver genes (MACF1 and MET) could be modulated by the m$^6$A alteration caused by METTL14 mutations. It was worthy of note that all of these genes had a higher mutation rate or copy number altered rate in iCCA than in pCCA. Thus, they were crucial potential driver genes in both iCCA and pCCA. However, they could be activated through different mechanisms. Although MACF1 and MET could serve as shared therapeutic targets for pCCA and iCCA, we should not ignore the fact that METTL14 could modulate m$^6$A modification of broader genes other than the two drivers, which may lead to unexpected drug resistance or side effects, and therefore, further investigation in pCCA patients is warranted. Another gene, RMB10, is an RNA binding protein and alternative splicing regulator frequently mutated in lung adenocarcinomas[38]. Similar to lung cancer, the majority of RBM10 mutations (63.6%) in CCA truncate the protein and may alter downstream splicing of specific genes[38,39]. Thus, the post-transcriptional events in CCA could be highly disturbed and further study may help determine new molecular subtypes and optimize therapeutic strategy in the clinical setting.

Our study also identified a number of crucial potential driver genes that were not reported in previous studies, involving multiple important pathways. MACF1 and AXIN1 were identified as SMGs of CCA in our study. They are involved in a complex that also contained CTNNB1, GSK3B, and APC, and can contribute to the activation of β-catenin/Wnt signaling pathways[28,40]. Consistently, the mutations in the two genes were mutually exclusive. CNA-driver MET belongs to the classic RTK-RAS signaling pathway, and its inhibitor has been widely used in other cancers[41,42]. Although the MET amplification has been reported in CCA, our study highlighted the presence of MET amplification is a genomic feature of pCCA, and reported with rare amplification frequency in Caucasian iCCA[43,44], suggesting that it may occur more frequently in Chinese patients. According to OncoKB, nearly 106 (30.5%) of all CCA patients carried

actionable alterations in the RTK-RAS signaling pathway. PIK3R1, phosphatidylinositol 3-kinase regulatory subunit alpha, is the predominant regulatory isoform of PI3K[45] and is frequently mutated in multiple cancers[46,47]. ATM served as the activator of the TP53 tumor suppressor protein and somatic ATM mutations or deletions that are commonly found in lymphoid malignancies, pancreatic cancer, and lung adenocarcinoma[48]. Notably, our results suggested that ATM could be affected by both truncating mutations and deletion, resulting in the bi-allelic inactivation. Several known tumor suppressors, such as TP53, SMAD4, PTEN, and CDKN2A, showed a similar bi-allelic loss pattern. Germline mutations in BRCA2, which encodes a BRCA-associated protein, were reported in familial CCA cases[49,50]. Our study also suggested that BRCA2 could be affected by somatic mutations. MLLT4, also known as AF6, commonly fused with KMT2A in acute myeloid leukemia. It has also been identified as a SMGs in breast cancers[51]. Few genomic alterations has ever been reported in NACC1, but it is activated in ovarian serous carcinomas and influences cell apoptosis, senescence, and cytokinesis in cancer cells[52,53]. Integrated whole-exome analysis of Asian CCAs identified significant CCA driver genes and emphasized the similarities and distinctions genomic characteristics between pCCA and iCCA. Importantly, our study also highlighted the effect of underlying post-transcriptional modification in CCA occurrence and development. These results provide a better understanding of the CCA mutational landscape that may drive improvements in clinical practice.

## Methods

**Clinical sample collection**. All primary cholangiocarcinoma and matched adjacent normal samples were obtained from the resected specimens of patients with pCCA or iCCA between 2010 and 2017 in The Affiliated Hospital of Nanjing Medical University (NMU). All samples were immediately frozen in liquid nitrogen and stored at −80 °C until DNA extraction. The use of clinical samples was approved by the Ethics Committee of The Affiliated Hospital of Nanjing Medical University. Written informed patient consent was obtained in accordance with regional regulation. The data of their clinicopathological features were anonymized and were shown in Supplementary Table 1. All tumor samples were confirmed by pathologists that there was a minimum tumor cellularity of 70% in all CCA specimens following histopathological review of H&E slides.

**Next-generation sequencing**. DNA extracted from frozen tumor specimens and adjacent normal tissues using QIAmp DNA mini kit (Qiagen) and quality were determined using Picogreen (Invitrogen) and further visually inspected by agarose gel electrophoresis. Library construction and whole-exome capture of genomic DNA were performed using the Roche NimbleGen SeqCap EZ Exome SR platform V3. The captured DNA was sequenced on an Illumina HiSeq X10 sequencing system, with 150-bp paired-end sequencing. The average sequence coverage was 91.9× and 97.6× for the pCCA and iCCA tumor samples and 89.7× and 94.6× for pCCA and iCCA adjacent normal tissues for whole-exome sequencing.

**Sequencing alignment and detection of somatic variants**. The quality score distribution of the reads was acquired with the FastQC package (http://www.bioinformatics.babraham.ac.uk/projects/fastqc).Burrows-Wheeler Aligner (BWA-MEM v0.7.15-r1140) was used to map the read sequences to the human reference genome (GRCh37) with the default parameters[54], and duplicates were marked and discarded using Picard (v1.70) (http://broadinstitute.github.io/picard). Then, the reads were subjected to local realignment and recalibration using the Genome Analysis Toolkit (GATK)(v4). For the 102 patients from Zou et al's study[17], which was approved by the Eastern Hepatobiliary Surgery Hospital Ethics Committee (Shanghai, China), we downloaded the raw FASTQ files from Short Read Archive (SRA) under the accession code SRP045202 and applied the same alignment pipeline mentioned above.

Somatic substitutions and indels were detected using the MuTect2 mode in GATK (v4) on the GRCh37 genome build following the best practices for somatic SNV/indel calling (https://software.broadinstitute.org/gatk/best-practices/). Briefly, the algorithms compared the tumor with the matched normal sample to exclude germline variants. Somatic mutations were excluded (1) if they were found in a panel of normal controls assembled from matched normal tissues, (2) if they were located in the segmental duplication region marked by the UCSC browser (http://genome.ucsc.edu/), or (3) if they were found in the 1000 Genomes Project (the Phase III integrated variant set release, across 2,504 samples) with the same mutation direction. Mutations (SNVs/indels) were annotated with the local versions of Oncotator (1.8.0)[55] according to GENCODE v19. In addition, the annotated somatic mutation R298H of METTL14 in tissue and cell lines were further confirmed by Sanger sequencing, the forward primers of METTL14 used in this study are 5'- CGAGGTAGGTAGACCACTTG-3', and reverse primer is 5'-TTCCAAATAGATGAAGGCGT-3'.

For integration analysis, we applied the somatic mutation data of 239 patients with cancer of biliary duct, including 44 pCCAs and 135 iCCAs, from BTCA-JP, International Cancer Genome Consortium (ICGC) project (https://icgc.org/icgc/cgp/91/420/1012366). The tissues and clinical information used in this study were obtained under informed consent and with the approval of the National Cancer Center (NCC) Institutional Review Board.

**Mutational burden and signature analysis**. The mutation counts were provided in Supplementary Table 2. The mutation burden was defined as the number of somatic mutations per mega base (/MB) in the callable regions, which was considered as the region of mapped genome with depth of at least 10×.

We converted all substitutions into a matrix (M) composed of 96 features comprising mutation counts for each mutation type (C > A, C > G, C > T, T > A, T > C and T > G) using each possible 5′ and 3′ context for all samples. R packages SignatureEstimation[56] was taken to estimate the proportion of 30 signatures from COSMIC (https://cancer.sanger.ac.uk/cosmic/signatures_v2). The signatures, percentage of which were more than one in pCCA or iCCA patients, were included in further analysis. (Supplementary Table 5).

**Identification of CCA potential driver genes and comparison between the iCCA and pCCA**. The IntOGen platform[57] and MutSig2CV (v3.11)[58] was used to identify SMGs among the somatic mutation data of all the patients from three databases. The IntOGen pipeline included two algorithms (OncodriveCLUST[59] and OncodriveFM[60]) that were designed to find genes with highly clustered mutations and non-randomly distributed functional mutations, respectively. MutSig2CV was used to find genes with a higher mutation rate than the calculated background mutation rate. Multiple testing correction (Benjamini–Hochberg FDR) was performed separately and genes with $q$ values ≤ 0.1 in any algorithm were reported as potential driver genes also mentioned as mut-drivers in this study (Supplementary Table 7). Then we divided the patients into two subgroups according to the CCA subtype, and comparison was conducted between mut-driver genes. Subtype-specific mut-drivers was defined as genes with log2 scaled mutation rate ratio of which were >1 between iCCA and pCCA. The specific Mutations were considered recurrent when occurred in (1) two or more NMU patients or (2) at least one NMU patient and other patient(s) from ICGC and/or Zou et al.'s dataset.

**Somatic copy number estimation**. ICGC samples were not included in this copy number analysis due to the inaccessibility of the origin FASTQ and/or BAM files of the peripheral blood from those subjects. The GATK best practices for somatic copy number alterations (CNAs) in exomes were used to detect CNAs from the whole-exome sequencing data (https://software.broadinstitute.org/gatk/best-practices/). The somatic copy number was estimated by ReCapSeg, which is implemented as part of GATK (v4). Briefly, the read counts for each of the exome targets were divided by the total number of reads to generate proportional coverage. A panel of normal (PON) controls was built using proportional coverage from normal samples. Each of the tumor samples was compared with the PON, after which tangent normalization was applied. Circular binary segmentation (CBS) was then applied to segment the normalized coverage profiles. Sex chromosomes (X and Y) were excluded from this analysis. In addition, to correct the effect of tumor purity on CNA, we also used TitanCNA to obtain the somatic copy number segment files adjusted tumor purity (Supplementary Table 3 and 6). CNAB was considered as the proportion of genome altered (amplified and/or deleted) divided by the length of the genome that all segments covered and the copy number segments with the absolute value of log2 scaled copy number ratio more than 0.2 was defined as altered segments (>0.2 for amplification and <−0.2 as deletion)[61].

**Highly amplified/deleted regions identification**. Copy number segments of patients (67 from NMU and 102 from Zou et al.'s studies) were used as input for GISTIC2[62] to identify significantly amplified/deleted regions with the default parameters. A default q value threshold (q < 0.25) was used to define identified frequently amplified/deleted regions. Cancer related genes, curated by COSMIC[63], located at frequently CNA focal regions were defined as potential CNA-driver genes.

**Pathway enrichment analysis**. Pathway enrichment analysis was performed on TCGA ten pan-cancer pathways listed in Sanchez-Vega et al' study[64] using a Fisher's exact test based on the hypergeometric distribution[65]. Briefly, it determines whether the fraction of genes of interest in the pathway is higher compared to the fraction of genes in the background.

**Annotation of genomic alterations upon clinical actionability**. The annotation was performed with OncoKB[66]. The actionable level was classified as level 1-4. Alterations with level 1 and 2 are Food and Drug Administration (FDA)-recognized or considered standard care biomarkers predictive of response to FDA-approved drugs in specific disease settings. Level 3 alterations are believed as predictive of response based on promising clinical data to targeted agents being tested in clinical trials whereas level 4 was the prediction of response on compelling biological evidence to targeted agents being tested in clinical trials. Clinically actionable sample frequency was calculated as the number of patients with at least one actionable alteration divided by all individuals counts in each subgroup.

**Cell culture and siRNA transfection**. CCA cell lines RBE and HCCC9810 were selected to establish the stable METTL14[R298H] and METTL14[wt] cells. Cells were maintained in DMEM medium (Gibco) with 10% fetal bovine serum (Biological Industries), penicillin/streptomycin 100 units/mL at 37 °C with 5% $CO_2$. All the siRNAs were ordered from GenePharma. Sequences for siRNA of MACF1 are: MACF1-homo-2920, 5'- CCUUAUCUCUUGGAACUAUU -3'; MACF1-homo-8572, 5'- GCAGAAAGCUCAGAAAUAUU -3'. Transfection was achieved by using Lipofectamine 3000 (Invitrogen) for the siRNA following manufacturer's protocols.

**Establishment of stable METTL14[R298H] and METTL14[wt] cells and functional assays**. To produce recombinant lentiviruses, vectors encoding empty control, METTL14[R298H] and METTL14[wt] gene were designed from GeneChem (Shanghai, China)[67,68]. The lentivirus transfection assay was performed in $2 \times 10^5$ cells using 5 μl/ml polybrene according to the manufacturer's protocol. Puromycin (10 μg/ml) was added for 7 days to select stable cells. Migration assays were performed in Transwells (Corning Inc., 8.0-μm pore size) and invasion assay was conducted using BD BioCoat Matrigel invasion chambers according to the manufacturer's instructions. Briefly, $2 \times 10^4$ cells in 300 μl serum-free medium were loaded into the upper chambers. Then, 500 μl medium supplemented with 20% fetal bovine serum was loaded into the lower chamber. Cells on the underside of the membrane were stained with crystal violet and counted under a microscope in three random fields after 48 h. For wound healing assay, cells were seeded in 6-well plates at 80–90% confluency in serum-free medium. A linear wound was performed in constant-diameter stripes using a sterile 200-μl pipette tip. After incubation for 0, 24, 48 h, photographs were taken and wound closures were evaluated and calculated to estimate the capacity of migration.

The cell proliferation was detected by EdU (5-ethynyl-2 0 -deoxyuridine) assay using Cell-Light EdU DNA Cell Proliferation Kit (RiboBio, Shanghai, China). Cells were seeded in each well of 6-well plates. Nucleic acids in all cells were stained with DAPI Dye. The Edu pulse-chase incorporation and cell proliferation rate were applied according to the manufacturer's instructions. For colony-forming assay, the transfected cells were replated and incubated for an additional 14 days at 37 °C to allow colony formation. Colonies immobilization were maintained with −20 °C methyl alcohol for 30 min and were stained with 0.5% crystal violet and counted. Cell proliferation rates were subsequently assessed using the cell counting kit-8 (CCK-8) (Dojindo, Tokyo, Japan) according to the manufacturer's instructions.

The cell apoptosis was monitored by Annexin V/PI apoptosis kit (Multisciences, Hangzhou, China). $3 \times 10^5$ cells seeded in 6-well plates were treated with 5 μl/ml 0.3% hydrogen peroxide. Cells were then stained with Annexin V/PI binding buffer for 5 minutes protected from light. Percentage of early, late apoptotic cells were quantified with FlowJo V10 according to the manufacturer's instructions (Supplementary Fig. 6).

**RNA N6-methyladenosine (m6A) quantification**. Total RNA was isolated using TRIzol (Invitrogen) according to the manufacturer's instructions and RNA quality was analyzed by NanoDrop. The EpiQuik™ m6A RNA Methylation Quantification Kit (Colorimetric) was used to assess the m6A content in total RNA. According to the manufacturer's instructions, 200 ng RNAs were an optimal amount on assay

wells. M6A standard control was added into the assay wells at different concentrations to determine the standard curve and then capture antibody, detection antibody, enhancer solution, and developer solution were added to assay wells respectively in a suitable diluted concentration. The m6A levels were quantified colorimetrically by the absorbance of each well at a wavelength of 450 nm, and then calculations were performed based on the standard curve.

**RNA m6A immunoprecipitation assay and m6A sequencing**. For m6A sequencing, total RNA was extracted using Trizol reagent (Invitrogen) following the manufacturer's procedure. The total RNA quality and quantity were analysis of Bioanalyzer 2100 and RNA 6000 Nano LabChip Kit (Agilent) with RIN number >7.0. Approximately more than 200 ug of total RNA was subjected to isolate Poly (A) mRNA with poly-T oligo attached magnetic beads (Invitrogen). Following purification, the poly(A) mRNA fractions are fragmented into ~100-nt-long oligonucleotides using divalent cations under elevated temperature. Then the cleaved RNA fragments were subjected to incubated for 2 h at 4 °C with m6A-specific antibody (No. 202003, Synaptic Systems) in IP buffer (50 mM Tris-HCl, 750 mM NaCl and 0.5% Igepal CA-630) supplemented with BSA. The mixture was then incubated with protein-A beads and eluted with elution buffer (1 × IP buffer and 6.7 mM m6A). Eluted RNA was precipitated by 75% ethanol. Eluted m6A-containing fragments (IP) and untreated input control fragments are converted to final cDNA library in accordance with a strand-specific library preparation by dUTP method. The average insert size for the paired-end libraries was ~100 ± 50 bp. And then we performed the paired-end 2 × 150 bp sequencing on an Illumina Novaseq™ 6000 platforms at the LC-BIO Bio-tech Ltd (Hangzhou, China) following the vendor's recommended protocol. Enrichment of m6A containing mRNA was analyzed by quantitative reverse-transcription polymerase chain reaction (qRT-PCR) using m6A RNA immunoprecipitation (MeRIP) method. In this assay, total RNA was chemically fragmented into 100 nucleotides or smaller RNA fragments, and m6A monoclonal antibody was used for magnetic immuno-precipitation. The above methods follow the standard protocol of the Magna MeRIP m6A Assay Kit (Merck Millipore).

**m6A sequencing analysis**. Reads were aligned to the reference genome (GRCh37) with HISAT2[69]. The m6A modification peaks were identified using R package exomePeak2[70] and a cutoff for controlling FDR < 0.05 was used to obtain high-confident m6A modification peaks. Then, we retrieved the raw read counts from both input and IP RNA-Seq bam files by Genomic Alignments and performed differential methylation analysis by DESeq2 and fitted the model as below:

Normalized counts ~ Design (e.g., METTL14wt vs control) + Experiments (IP/control) + Design × Experiments.

Candidate differential m6A peaks was selected with (1) Multiple testing correction P value ≤ 0.05 and (2) log2 Fold Change > 1 between METTL14wt and control cells or log2 Fold Change < 0 between METTL14R298H and METTL14wt cells.

**Measurement of RNA lifetime**. RBE and HCCC9810 cells were seeded in 6-well plates at 80% confluency. After 24 h, actinomycin D (MedChemExpress) was added to 1 mg/ml at 6 h, 3 h, and 0 h before trypsinization and collection. The total RNA was purified according to the procedure mentioned in this study. The degradation rate of RNA (k) was estimated by plotting Nt/N0 against time follow the standard equation mentioned by Liu et al.[71].

**Subcellular fractionation, Western blotting and immunohistochemistry staining**. For protein expression analysis, cells and tissues were lysed using Radioimmunoprecipitation (RIPA) lysis buffer (KeyGen Biotech) containing protease and phosphatase inhibitors (Roche) to obtain the total protein, and protein concentration were estimated using the BCA protein estimation assay (Thermo Scientific). Subcellular fractionation (including nuclear, membrane, and cytoplasm) was performed using cell fractionation kit (Cell Signaling Technology). Equal amounts of total protein were separated by SDS-PAGE and subsequently transferred to nitrocellulose membrane. Antibodies used were: anti-METTL14 (Novus), anti-MACF1 (Proteintech), anti-GAPDH (Abcam), anti-Lamin B1 (Proteintech), anti-β-catenin (Abcam), anti-E-cadherin (Cell Signaling Technology), anti-N-cadherin (Cell Signaling Technology), anti-α-Tubulin (Cell Signaling Technology), anti-PCNA (Abcam), anti-Cyclin D1 (Cell Signaling Technology), and ATP1A1 (Cell Signaling Technology).

Immunohistochemical staining for METTL14 was performed on the cholangiocarcinoma (including iCCA and pCCA) tissue array using anti-METTL14 (Novus). After 4% paraformaldehyde fixation, the sections were deparaffinized. Antigens of the slides were retrieved by heating for 30 min in citrate buffer, pH 6.0. The slides were labeled with primary antibody in a blocking solution at 4 °C overnight, following by counterstained with hematoxylin. Light microscopy (Nikon) was used to acquire the images, and NIS-Elements v4.0 software was used to quantify the staining (Nikon). All antibodies used in this study were listed in the reporting summary.

**RNA isolation, quantitative real-time PCR and RNA immunoprecipitation**. Total RNAs were isolated from specimens or cell lines using TRIzol (Invitrogen)

following the manufacturer's protocol. RNA quality was analyzed by NanoDrop. The high capacity cDNA transcription kit (Vazyme) qRT-PCR was performed by the Thermal Cycler Dice Detection System using SYBR Green PCR Mix (Vazyme). The forward primers of METTL14 used in this study are 5'- AGTGCCGACAGCATT GGTG', and reverse primer is 5'- GGAGCAGAGGTATCATAGGAAGC-3'. The forward primers of MACF1 are 5'- CGGAGTGAGCGATCTACAGG -3', and reverse primer is 5'-TCATCAGCGACTCTGACCACA -3'. All data were normalized to the housekeeping gene GAPDH. RNA immunoprecipitation was performed in CCA cell lines using RNA Immunoprecipitation Kit (Geneseed, China) following the manufacturer's instructions. Magnetic beads pre-coated with 5 mg normal antibodies against METTL14 (Proteintech, #26158-1-AP) or IgG (Cell Signaling Technology) were incubated with sufficient cell lysates (more than 2 × 10^7 cells per sample) at 4 °C overnight. The interested RNAs were purified and detected by RT-qPCR.

**Evaluation of immunostaining**. Hematoxylin and eosin (H&E) staining was performed routinely. Tissue microarrays (TMA) containing 114 cases of CCA specimens and corresponding adjacent normal tissues, which were obtained from The Affiliated Hospital of Nanjing Medical University. Scoring was conducted based on the percentage of positive-staining cells: 0–5% scored 0, 6–35% scored 1, 36–70% scored 2, and more than 70% scored 3; and staining intensity: no staining scored 0, weakly staining scored 1, moderately staining scored 2 and strongly staining scored 3. The final score was calculated using the percentage score × staining intensity score as follows: "-" for a score of 0-1, "+" for a score of 2-3, "++" for a score of 4-6 and "+++" for a score of >6. Low expression was defined as CCA specimens' score < corresponding adjacent normal tissues' score, and high expression was defined as CCA specimens' score ≥ corresponding adjacent normal tissues' score. These scores were determined independently by two senior pathologists in a blinded manner. Specifically, positive staining in the hepatic tissue was excluded for scoring.

**Immunofluorescence analysis**. Cells were washed with phosphate-buffered saline, fixed in 4% paraformaldehyde for 10 min and permeabilized with 0.25% Triton X-100 in phosphate-buffered saline for 5 min, followed by 1 h incubation with primary antibodies, METTL14 (Novus), MACF1 (Abcam) and then incubation with IgG (Santa Cruz). The coverslips were counterstained with 4,6-diamidino-2-phenylindole (Invitrogen) and imaged with a confocal laser-scanning microscope (Olympus FV1000, Olympus).

**Animal experiments**. All the animal studies were approved by the Institutional Animal Care and Use Committee of Nanjing Medical University and conducted according to protocols approved by the Ethical Committee of Nanjing Medical university. For the subcutaneous tumor growth assay, 4-week-old BALB/c nude male mice were purchased from Animal Core Facility of Nanjing Medical University (Nanjing, China). Suspended in 100 μL serum-free PBS, 2 × 10^6 RBE cells transfected with lentivirus expressing METTL14R298H and METTL14wt were subcutaneously injected into the flanks of the mice. All mice were sacrificed four weeks later. Tumor volume was measured as follows: volume = length × width^2 × 0.5. The maximum diameter of the tumor does not exceed 1.5 cm. For the lung metastasis experiment, 5 × 10^6 of the indicated cells were suspended in 0.1 mL PBS and injected into the lateral tail vein of 6-week-old male nude mice. Metastases were detected using the IVIS® Lumina XR Series III. All mice were sacrificed after inoculation of 6 weeks, then the metastatic nodes in the lungs were examined by necropsy and counted. Animal experiments were per-formed with the approval of Animal center of Nanjing medical university and use committees.

**Statistical analysis and figures**. The median was used if multiple samples from the same tissues were sequenced. All statistical tests were performed using a Wilcoxon rank-sum test for continuous data. Fisher's exact test was used to assess differences in the count data. Multiple testing corrections were performed where necessary using the Benjamini-Hochberg method. All reported P values were two-sided. Mutational lolliplots were generated by ProteinPaint (https://proteinpaint.stjude.org)[72]. Other figures were generated using the, R with package ggplot2 [Wickham H (2016). ggplot2: Elegant Graphics for Data Analysis. Springer-Verlag New York. ISBN 978-3-319-24277-4,] and package RColorBrewer [https://cran.r-project.org/web/packages/RColorBrewer/index.html].

**Reporting summary**. Further information on research design is available in the Nature Research Reporting Summary linked to this article.

## Data availability

The WES (accession number: HRA001570) and RNA m6A sequencing (accession number: HRA001826) raw data of NMU patients with pCCA or iCCA have been deposited in Genome Sequence Archive (GSA) for human hosted by China National Center for Bioinformation (CNCB), and can be accessed publicly. We have got approval from MOST to share the sequencing data in this study (Record Number: 2022BAT1241).

The publicly available WES raw data obtained from Zou et al's study were downloaded from Short Read Archive (SRA) under the accession code SRP045202. The raw data of

BTCA-JP were obtained from International Cancer Genome Consortium (ICGC) project [https://dcc.icgc.org/projects/BTCA-JP]. All other data is available within the Article, Supplementary Information or Source Data included with the manuscript.

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

## Acknowledgements

This study was supported by Natural Science Foundation of Jiangsu Province, China (BK20171077 to Xiangcheng Li), Key research and development program of Jiangsu Province (BE2016789 to Xiangcheng Li), National Natural Science Foundation of China (NSFC) (81700572 to Changxian Li, 81670570 to Xiangcheng Li), National Science and Technology Major Project of China (2017ZX10203207-004-004 to Xiangcheng Li).

## Author contributions

Y.Z.: data acquisition and analysis, manuscript drafting. Z.M.: statistical analysis and interpretation of data, manuscript drafting. C.L.: data acquisition and analysis, manuscript revision. C.W.: critical revision of the manuscript and study concept and design. W.J.: data acquisition. J.C.: data acquisition. S.H: data acquisition. Z.L.: data acquisition. Z.S.: data acquisition. Y.W.: data acquisition. H.W: data acquisition. C.J.: data acquisition. D.W.: data acquisition. X.W.: administrative, technical, or material support. HB.S.: administrative, technical, or material support. XH.W.: administrative, technical, or material support. Z.H.:study supervision, critical revision of the manuscript and study concept and design. X.L.:critical revision of the manuscript, data analysis and study concept and design.

## Competing interests

The authors declare no competing interests.
