## [Peer Review File · Nature Communications]

The genomic landscape of cholangiocarcinoma reveals the disruption of post-transcriptional modifiersREVIEWER COMMENTS

Reviewer #1 (Remarks to the Author): Expert in m6A and epitranscriptomics

This manuscript reported genomic heterogeneity and a potential role of m6A RNA methylation in cholangiocarcinoma (CCA) using exome-sequencing and analyzing existing data, followed by validation of the role of METTL14 using in vitro models. They showed that METTL14 is mutated in a small portion of CCA and may serve as a potential tumor suppressor in ACC. One potential target for METTL14 is MACF1, which may regulate CCA traits through the beta-catenin pathway. METTL14 R298H mutation seems to lose METTL14 function. The manuscript showed a potential new mechanism in CCA pathogenesis and the role of METTL14 mutation. However, multiple weaknesses are noted, including over-interpretation of the data and lack of in vivo and mechanistic evidence to support the conclusion of the manuscript.

Specific comments:

1. The manuscript overly used the phrase "driver" in general, in many places. For example, Line 142, 148, 214, 248, 339 for METTL14, and Line 277, 345, and L361 for MACF1. However, there are no data from this manuscript to support such a statement. Do mice with deletion of either gene drive CCA development?
2. Exome sequencing identified mutations of multiple genes. What is the rationale that the authors focus on METTL14? How about MACF1 or RBM10? Are they also important for CCA tumorigenicity?
3. Only IF was used to determine the effect of WT and R298H on MACF1. More quantitative methods need to be used to address this question, such as Western blotting. Although the authors indicated the size of MACF1 can be a challenge, it can be done with optimized methods and larger proteins have been analyzed by Western blotting in the past.
4. The authors showed that the potential importance of MACF1 in mediating METTL14 R298H function by knocking down MACF1 in cells overexpressing METTL14 R298H. Further experiments are required to support the conclusion, such as whether overexpression of MACF1 will prevent the effect of WT METTL14.
5. Although m6A motif has been illustrated in Supplementary Figure 4A, the m6A peaks across the MACF1 transcript is not shown. What are the sites for m6A modification in MACF1?
6. It is not shown how METTL14 regulates MACF1, and how R298H mutant loses its function. Does METTL14 bind to MACF1 transcript, while R298H does not?
7. Supplementary Figure 4G: cellular fractionation should be used to assess the role of METTL14/MACF1 on beta-catenin nuclear localization.
8. Most in vitro experiments only used one cell line, which reduces the rigor of the study. For key experiments, at least a second cell line needs to be used to determine whether the effect observed is cell line dependent or validated.
9. The study lacks in vivo data to support the importance of METTL14, R298H, or MACF1 in ACC tumorigenicity. Only in vitro model is used in investigating the role of METTL14 and R298H mutant. This is a major weakness.
10. Does METTL14 R298 mutation or MACF1 mutation affect proliferation, aggressiveness, or prognosis of ACC in human samples?

Minor comments:

1. Line 57: "western" needs to be revised.
2. Line 232: Citation 28, 29 seems not to support the statement and needs to be revised.

Reviewer #2 (Remarks to the Author): Expert in cholangiocarcinoma

The manuscript NCOMMS-20-48057 reports on exome-sequencing of Asian patients with iCCA and pCCA. The data demonstrate differences in the mutational landscape between iCCA and pCCA. More importantly the data indicate that mutations of METT14 and RMB10 are relatively unique to pCCA and disrupt RNA modifications. These post-transcriptional modifications affect expression of MACF1, a new driver gene. The genetic studies are well done and rigorous. The results are

important to our understanding of pCCA biology. These genetic aberrations are further explored by functional studies in CCA cell lines which are compelling and help define the impact of the genetic studies. I only have modest comments.

MAJOR COMMENTS:

1. 1q21 amplification was identified as S100A7 amplifications but usually this amplification contains several loci including BC19, MCL1, and ARNT. The authors suggest that this is only a S100A7 amplification. Please clarify.
2. The functional studies examine cell growth, invasion and colony formation. These studies should also include studies of apoptosis, especially sensitivity to apoptosis with a pro-apoptotic stimulus.
3. An in vivo correlate such as a transfected xenograft, PDX, or human organoid models would greatly enhance the manuscript.

MINOR COMMENTS:

1. Page 4, line 63. "additional 44 pCCA and 237 CCA..." what was the anatomic subtype of the 237 CCA patients?
2. The amplification of MET has been described before in CCA; please see the early work of Al Sirica, Virginia.
3. Page 4, line 270 starts with "y" this seems to be extraneous consonant.

Reviewer #3 (Remarks to the Author): Expert in cholangiocarcinoma genomics

This paper reported exome sequencing of 87 perihilar cholangiocarcinoma (pCCA) and 261 intrahepatic cholangiocarcinoma (iCCA) including previously reported data set by other groups. By comparing pCCA and iCCA, they reported that iCCA harbored more total mutation burden and copy number alterations, and the latter was associated with poorer prognosis. Among newly identified driver genes, METLL14 is a pCCA-enriched driver and its loss-of-function mutant suppressed m6A modification of MACF1, an iCCA-enriched driver. They suggest that aberrant post-transcriptional modification plays a role in CCA development.

This study is generally well performed and conveys interesting observations such as reporting transcriptional modifiers as novel CCA driver genes. However, lacking transcriptomic and epigenetic data diminished the significance of this research. It would be more interesting if the author could determine whether cases with alterations of these transcriptional modifiers show characteristic expression/epigenetic subtypes.

1. Significance difference of the non-synonymous mutation number between pCCA and iCCA
This result is interesting but requires additional analysis. Because the ratio of non-synonymous mutation/synonymous mutation (dN/dS) would be 0.44 (0.61/1.39) in iCCA and 0.31 (0.47/1.53) in pCCA, a selective pressure on somatic mutations is stronger in iCCA. First of all, clonal mutations may be more strongly selected than subclonal mutations. Therefore, it is required to determine whether the tumor purity and the ratio of clonal/subclonal non-synonymous mutations are comparable between iCCA and pCCA. Secondly, it may be possible that negative selection of neo-antigenic mutations by the host immune system would be different in the two subtypes. Is the total number of putative neo-antigenic mutations different between iCCA and pCCA?

2. Mutation signature

It is odd to see signature 7 (by UV exposure) in internal organ cancers. Did this signature in this cohort show a characteristic strand bias?

3. CNA (Figure 3)

Copy number should be adjusted by the tumor purity. The method by this study seems to rely solely on the read count and not considering the tumor purity. The author should use other algorithms considering tumor purity for evaluating CNA.

(Fig3B) It is hard to discriminate splice site mutation and copy loss because both are indicated by pale-blue. The author should re-make the better presentation.

(Fig3D) It is not surprising that RAS-RTK alterations are generally mutually exclusive. Better to move this panel to the supplementary data.

4. METTL14

- 1) Do cases with METTL14 mutations have LOH of the wild allele? It is unclear whether this mutation is heterozygous (maybe dominant negative) or two-hit inactivated (classical TSG).
- 2) In Fig 4E/4F/4G, is the difference between NC and R298H significantly different? If R298H functions as a dominant-negative form, R298H should inhibit endogenous METTL14 function. How about the METTL14 mutation status in the two CCC cell lines?
- 3) Fig 4H and Fig 5G, there is a typo ("clony" should be colony)
- 4) How about the METTL14 mutant effect on the MET transcript?

5. MACF1

- 1) Do cases with low METTL14 expression (Fig 4B) show higher MACF1 in the primary tumor cohort?
- 2) METTL14 mutant effect on b-catenin signaling: In Fig 5K, the amount of nuclear b-catenin is comparable between NC and R298H expressing clones (lanes 4 and 6), but the expression of cyclinD1 is more decreased in R298H expressing clones. It seems that METTL14-MACF1-WNT would be more complicated than the author proposed. How to settle this inconsistency?

Dear reviewers,

We would like to express our sincere appreciation for your careful reading and invaluable comments to improve our manuscript. We have tried our best to address the issues raised by the reviewers. The amendments made are mentioned below with reference to appropriate paragraphs and sections of the revised manuscript.

Point-by-point responses to reviewer concerns:

Response to Reviewer #1:

[**Comment1**] This manuscript reported genomic heterogeneity and a potential role of m6A RNA methylation in cholangiocarcinoma (CCA) using exome-sequencing and analyzing existing data, followed by validation of the role of *METTL14* using in vitro models. They showed that *METTL14* is mutated in a small portion of CCA and may serve as a potential tumor suppressor in ACC. One potential target for *METTL14* is *MACF1*, which may regulate CCA traits through the beta-catenin pathway. *METTL14* R298H mutation seems to lose *METTL14* function. The manuscript showed a potential new mechanism in CCA pathogenesis and the role of *METTL14* mutation. However, multiple weaknesses are noted, including overinterpretation of the data and lack of in vivo and mechanistic evidence to support the conclusion of the manuscript.

[**Answer**] Thank you for your interest in our research. For your constructive suggestions regarding our research, we will give detailed revisions in the answers below.

[**Comment2**] The manuscript overly used the phrase “driver” in general, in many places. For example, Line 142, 148, 214, 248, 339 for *METTL14*, and Line 277, 345, and L361 for *MACF1*. However, there are no data from this manuscript to support such a statement. Do mice with deletion of either gene drive CCA development?

[**Answer**] We agree with the reviewer’s comment. The significantly mutated and copy number altered genes identified in CCA patients should be considered as potential driver genes. So, we modified the word “driver” to “potential driver” in the description of the genomic landscape of CCA patients. For example, we modified the description of LINE 12 to “..., post-transcriptional modification-related novel **potential driver** genes *METTL14* and *RBM10*.”. In

this revised manuscript, we have added *in vivo* and *in vitro* experiments and found that *METTL14*^{R298H} mutation reduced the *METTL14*^{wt} tumor-suppressing effect, further supporting that *METTL14* may act as a driver gene of CCA. Meanwhile, *in vitro* experiments suggested that *MACF1* was a significant oncogenic gene in CCA, which also supported *MACF1* may as a driver gene in CCA. Therefore, “potential driver gene” was used to demonstrate *METTL14* and *MACF1* in “new potential driver genes” section of the revised manuscript. After the significant function of *METTL14* and *MACF1* in cholangiocarcinoma was revealed by the basic experimental in our study, “driver gene” was used to describe *METTL14* and *MACF1*.

[Comment3] Exome sequencing identified mutations of multiple genes. What is the rationale that the authors focus on *METTL14*? How about *MACF1* or *RBM10*? Are they also important for CCA tumorigenicity?

[Answer] Post-transcriptional modification genes *RBM10* (3.16%, 11/348) and *METTL14* (0.86%, 3/348) act as well-known alternative splicing regulators and m⁶A writers, respectively. As we observed a position recurrent mutation in *METTL14* in two patients and the variants were also predicted as potentially deleterious variants, we mainly focused on *METTL14* in the original manuscripts.

As suggested by the reviewer, we further investigated the role of *RBM10* in this revision. We first performed qRT-PCR to demonstrate the expression of *RBM10* in CCA and matched adjacent normal tissues. The result showed that *RBM10* was downregulated in tumors (Fig.Reponse1A). Downregulation of *RBM10* in immunohistochemistry staining on tissue microarray displayed a significant association with poor cancer-specific survival in CCA (Fig.reponse1B). Among all the non-synonymous mutations of *RBM10*, we selected the mutation C761Y for further experiments, which was predicted with the most functional impact of amino acid substitution with MutationAssessor (score 0.89601). We successfully constructed *RBM10*^{wt} and *RBM10*^{C761Y} mutant cell lines (Fig.reponse1C). Overexpression of wild-type *RBM10* affected cell proliferation and metastasis significantly, and the *RBM10*^{C761Y} mutation could reduce the tumor-suppressing effect of *RBM10*^{wt} in CCA. For *MACF1*, our research showed that *MACF1* played an important role in CCA development (Fig.5, Supplementary Fig.5). The above work will be published in our subsequent studies.

Figure.Response1 *RBM10*^{C761Y} mutation reduced the *RBM10*^{wt} tumor-suppressing effect in CCA. (A) Downregulated *RBM10* mRNA expression was detected in 65 pairs of CCA tissues as well as the adjacent normal tissues by qRT-PCR. (B) *RBM10* downregulation in CCA tissues was associated with shorter cancer-specific survival in CCA patients. (C) Stable *RBM10*^{wt} and *RBM10*^{C761Y} Cells were screened by western blot. (D) Colony formation assay of CCA cells with *RBM10*^{wt}, *RBM10*^{C761Y}, or negative control. (E) Representative images (left) and quantification (right) of transwell migration and invasion assays in CCA cells with *RBM10*^{wt}, *RBM10*^{C761Y}, or negative control.

[Comment4] Only IF was used to determine the effect of WT and R298H on *MACF1*. More quantitative methods need to be used to address this question, such as Western blotting. Although the authors indicated the size of *MACF1* can be a challenge, it can be done with optimized methods and larger proteins have been analyzed by Western blotting in the past.

[Answer] Thank you for pointing this out. We set the power supply to 250 mA for 10h at 4 °C in the process of electrotransfer, and succeeded to detect the stripe of *MACF1* on the PVDF membrane. We added the western blotting results of *MACF1* in the revised manuscript (Fig.5G, Supplementary Fig.4K).

[Comment5] The authors showed that the potential importance of *MACF1* in mediating *METTL14* R298H function by knocking down *MACF1* in cells overexpressing *METTL14* R298H. Further experiments are required to support the conclusion, such as whether overexpression of *MACF1* will prevent the effect of WT *METTL14*.

[Answer] We thank the Reviewer for pointing this out. *MACF1*, belonging to the cytoskeletal linker protein family with a molecular weight of approximately 600 kD, the establishment of a stable overexpression system of this large protein cannot be carried out by the lentivirus system. According to the other study in exploring the functional role of *MACF1* in cell lines¹, siRNA was the most commonly used method for the investigation of *MACF1*. To further support the relationship between *METTL14* and *MACF1*, we validated *MACF1* upregulation in CCA using qRT-PCR, and noticed that lower *METTL14* mRNA levels exhibited stronger *MACF1* expression in mRNA level in our revised manuscript (Fig.5L).

[Comment6] Although m6A motif has been illustrated in Supplementary Figure 4A, the m6A peaks across the *MACF1* transcript is not shown. What are the sites for m6A modification in *MACF1*?

[Answer] Thank you for your suggestion. We identified two m6a modification peaks on the 50-54th exons (peak2: chr1:39847798-39889746) and the 92nd exon (peak1: chr1:39952661-39952849) of *MACF1* and presented the results in Supplementary Fig. 4C.

[Comment7] It is not shown how *METTL14* regulates *MACF1*, and how R298H mutant loses its function. Does *METTL14* binds to *MACF1* transcript, while R298H does not?

[Answer] We thank the Reviewer for this insightful suggestion. According to the previous study, p.R298 of *METTL14* belongs to the groove between *METTL3* and *METTL14* which might be responsible for RNA binding, hence we supposed that *METTL14*^{R298H} may regulate

the enrichment of *MACF1* mRNA. Using RNA immunoprecipitation, we found that *METTL14*-specific antibody obversely enriched *MACF1* mRNA in *METTL14^{wt}* compared to the *METTL14^{R298H}* group (Fig.5E, Supplementary Fig.4F).

[**Comment8**] Supplementary Figure 4G: cellular fractionation should be used to assess the role of *METTL14/MACF1* on beta-catenin nuclear localization.

[**Answer**] We thank the Reviewer for bringing up this important point. To further explore the localization of beta-catenin in cholangiocarcinoma cells (RBE and HCCC9810) in detail, subcellular fractionation (including nuclear, membrane, and cytoplasm) was performed using a cell fractionation kit (Cell Signaling Technology). Equal amounts of total protein were separated by SDS-PAGE and subsequently transferred to nitrocellulose membrane (Fig.5N, Supplementary Fig.5G). The cellular fractionation showed that the increase of nuclear β -catenin was correlated with the expression of *METTL14^{R298H}* rather than *METTL14^{wt}*.

[**Comment9**] Most in vitro experiments only used one cell line, which reduces the rigor of the study. For key experiments, at least a second cell line needs to be used to determine whether the effect observed is cell line dependent or validated.

[**Answer**] We thank the reviewer for this constructive suggestion. In the revised manuscript, we performed the *in vitro* experiments and main experiments using another cholangiocarcinoma cell line, HCCC9810 (Supplementary Fig.3, Supplementary Fig.4 & Supplementary Fig.5), to validate our results and conclusions.

[**Comment10**] The study lacks in vivo data to support the importance of *METTL14*, R298H, or *MACF1* in ACC tumorigenicity. Only in vitro model is used in investigating the role of *METTL14* and R298H mutant. This is a major weakness.

[**Answer**] We thank the Reviewer for this constructive suggestion. The in vivo experiments can significantly improve the understanding of the function of *METTL14^{R298H}* mutations. To support our previous findings of *METTL14^{wt}/METTL14^{R298H}* *in vitro*, we have added *in vivo* experiments (subcutaneous tumor growth assay and lung metastasis experiment, refer to Fig4 J&K) and suggested that *METTL14* may act as a driver gene in CCA development.

[**Comment11**] Does *METTL14* R298 mutation or *MACF1* mutation affect proliferation, aggressiveness, or prognosis of ACC in human samples?

[**Answer**] We thank the reviewer for this constructive suggestion. Among the three patients with mutation *METTL14* R298, two were from our xMU cohort and we followed up on the disease status of these patients and found that cancer recurred (recurrence time: 299 and 722 days). Among four patients from xMU with *MACF1* mutation, one was recurred, one was possibly lung metastasis and one was intrahepatic metastasis (recurrence time: 214, 242 and 317 days). Because only two mutations of *METTL14* and six mutations of *MACF1* were observed in CCA patients, the prognostic effect of *METTL14* and *MACF1* can hardly be evaluated in the cohort.

[**Comment12**] Line 57: “western” needs to be revised.

[**Answer**] We have revised this mistake in the revised manuscript.

[**Comment13**] Line 232: Citation 28, 29 seems not to support the statement and needs to be revised.

[**Answer**] We have replaced these citations in the revised manuscript (with tracked changes: Line 236: Citation 27, 29; without tracked changes: Line 232: Citation 27, 29).

Reference:

1. Chen, H.J. *et al.* The role of microtubule actin cross-linking factor 1 (MACF1) in the Wnt signaling pathway. *Genes Dev* **20**, 1933-1945 (2006).

Response to Reviewer #2:

[**Comment1**] The manuscript NCOMMS-20-48057 reports on exome-sequencing of Asian patients with iCCA and pCCA. The data demonstrate differences in the mutational landscape between iCCA and pCCA. More importantly the data indicate that mutations of *METTL4* and *RMB10* are relatively unique to pCCA and disrupt RNA modifications. These post-transcriptional modifications affect expression of *MACF1*, a new driver gene. The genetic studies are well done and rigorous. The results are important to our understanding of pCCA biology. These genetic aberrations are further explored by functional studies in CCA cell lines which are compelling and help define the impact of the genetic studies. I only have modest comments.

[**Answer**] We thank the reviewer's positive comments.

[**Comment2**] 1q21 amplification was identified as *S100A7* amplifications but usually this amplification contains several loci including *BC19*, *MCL1*, and *ARNT*. The authors suggest that this is only a *S100A7* amplification. Please clarify.

[**Answer**] We thank the reviewer for pointing this out. We check the amplification peak in 1q21 and found that the peak (chr1:151630448-153750341) did not affect *BC19*, *MCL1*, and *ARNT*. Thus, we presented the region with the affected oncogene *S100A7*.

[**Comment3**] The functional studies examine cell growth, invasion and colony formation. These studies should also include studies of apoptosis, especially sensitivity to apoptosis with a pro-apoptotic stimulus.

[**Answer**] Thanks for the reviewer's suggestion. Apoptosis-related *in vitro* experiments were conducted in the revised manuscript, the results revealed that overexpression of *METTL4*^{wt} significantly affected cell apoptosis, and *METTL4*^{R298H} remarkably reversed the ability of *METTL4*^{wt} to promote apoptosis in CCA cells (Fig.5I and Supplementary Fig.4A).

[**Comment4**] An *in vivo* correlate such as a transfected xenograft, PDX, or human organoid models would greatly enhance the manuscript.

[**Answer**] We thank the reviewer for this insightful suggestion. There is no doubt that these methods can significantly improve the understanding of the function of *METTL4*^{R298H}

mutations. However, it is difficult to recruit patients with *METTL14*^{R298H} mutation because of the relatively low mutation rate. We will devote ourselves to solving this problem in future research. To support our previous findings of *METTL14*^{wt}/*METTL14*^{R298H} *in vitro*, we have added *in vivo* experiments (subcutaneous tumor growth assay and lung metastasis experiment, refer to Fig4 J&K) to support that *METTL14* may act as a driver gene in CCA development.

[**Comment5**] Page 4, line 63. “Additional 44 pecan and 237 CCA...” what was the anatomic subtype of the 237 CCA patients?

[**Answer**] We apologize for the mistake in our original manuscript. We have revised this mistake in the revised manuscript.

[**Comment6**] The amplification of MET has been described before in CCA; please see the early work of Al Sirica, Virginia.

[**Answer**] We thank the reviewer for pointing this out. Al Sirica’s work was using immunohistochemistry to demonstrate that overexpression of *MET* is a common feature of neoplastic biliary epithelial cells in liver and suggested that *MET* may be playing an important role in CCA. In our study, we highlighted the presence of *MET* amplification is a genomic feature of pCCA based on the Al Sirica’s work. We have revised the relevant descriptions (with tracked changes: Line 403–405; without tracked changes:Line 394-396) and cited the reference in the revised manuscript (with tracked changes:Line 405: Citation 46; without tracked changes: Line 396: Citation 46).

[**Comment7**] Page 4, line 270 starts with “y” this seems to be extraneous consonant.

[**Answer**] We have revised this word in our revised manuscript.

Response to Reviewer #3:

[**Comment1**] This paper reported exome sequencing of 87 perihilar cholangiocarcinoma (pCCA) and 261 intrahepatic cholangiocarcinoma (iCCA) including previously reported data set by other groups. By comparing pCCA and iCCA, they reported that iCCA harbored more total mutation burden and copy number alterations, and the latter was associated with poorer prognosis. Among newly identified driver genes, *METLL14* is a pCCA-enriched driver and its loss-of-function mutant suppressed m6A modification of *MACFI*, an iCCA-enriched driver. They suggest that aberrant post-transcriptional modification plays a role in CCA development. This study is generally well performed and conveys interesting observations such as reporting transcriptional modifiers as novel CCA driver genes. However, lacking transcriptomic and epigenetic data diminished the significance of this research. It would be more interesting if the author could determine whether cases with alterations of these transcriptional modifiers show characteristic expression/epigenetic subtypes.

[**Answer**] We would like to express our sincere appreciation for your careful reading and valuable comments to improve our manuscript. At the start stage of this study, we had tried to conduct transcriptomic sequencing and epigenetic data of CCA patient's tissues. However, due to the difficulty in obtaining cholangiocarcinoma tumor tissue and corresponding normal bile duct epithelial tissue, the volume of surgical samples is limited. Therefore, after exome sequencing, the most of remaining tissue samples are not enough to meet the experimental requirements of transcriptomic sequencing and epigenetic-related experiments. We have admitted that lacking transcriptomic and epigenetic data is a limitation of this study.

[**Comment2**] Significance difference of the non-synonymous mutation number between pCCA and iCCA This result is interesting but requires additional analysis. Because the ratio of nonsynonymous mutation/synonymous mutation (dN/dS) would be 0.44 (0.61/1.39) in iCCA and 0.31 (0.47/1.53) in pCCA, a selective pressure on somatic mutations is stronger in iCCA. First of all, clonal mutations may be more strongly selected than subclonal mutations. Therefore, it is required to determine whether the tumor purity and the ratio of clonal/subclonal non-synonymous mutations are comparable between iCCA and pCCA. Secondly, it may be possible that negative selection of neo-antigenic mutations by the host immune system would

be different in the two subtypes. Is the total number of putative neo-antigenic mutations different between iCCA and pCCA?

[Answer] We thank the reviewer for this insightful suggestion. We detected the purity using TitanCNA v1.23.1 and found that there is no significant difference between iCCA and pCCA (Wilcoxon rank-sum test P -value = 0.15, Figure.Response 2A). The clonal/subclonal mutation ratio was also found with no significant difference (Wilcoxon rank-sum test P -value = 0.56, Figure.Response 2B). Next, we detected the possible neoantigens burden between the two subtypes with NeoPredPipe v1.1 and there is no significant difference in the total burden of neoantigens between the two groups (Wilcoxon rank-sum test P -value = 0.14, Figure.Response 2C).

Figure.Response 2. The tumor purity, clonal/subclonal mutation ratio and neoantigen burden of CCA patients

[**Comment3**] It is odd to see signature 7 (by UV exposure) in internal organ cancers. Did this signature in this cohort show a characteristic strand bias?

[**Answer**] We thank the reviewer for this insightful suggestion. In our study, we extracted the mutation signatures based on the reference of COSMIC signatures and the approach assigns all signatures from the reference list to all samples(31278357). Because the proportion of Signature 7 mutations was relatively low (2.17% in iCCA, with tracked changes:LINE 128; without tracked changes:Line 126), we could not completely ascertain that the signature was the UV-related mutation. Thus, we included the signature into the "other" group in Fig.1 in the revised version.

[**Comment4**] Copy number should be adjusted by the tumor purity. The method by this study seems to rely solely on the read count and not considering the tumor purity. The author should use other algorithms considering tumor purity for evaluating CNA. (Fig3B) It is hard to discriminate splice site mutation and copy loss because both are indicated by pale-blue. The author should re-make the better presentation. (Fig3D) It is not surprising that RAS-RTK alterations are generally mutually exclusive. Better to move this panel to the supplementary data.

[**Answer**] Thanks for your suggestion. We try to get the somatic CNA segments adjusted for the purity by TitanCNA and then performed the GISTIC2 analysis. For the copy number alteration burden (CNAB) analysis, iCCA also had a significantly higher CNAB than pCCA, no matter the amplification or deletion. We added these results to the Supplementary Table 3 and described it in the manuscript (with tracked changes:LINE 101; without tracked changes:Line 99: “The CNAB based on purity-adjusted copy number in iCCA was also significantly higher than it in pCCA (All burden: Wilcoxon rank-sum test $P = 8.0 \times 10^{-4}$; Amplification burden: Wilcoxon rank-sum test $P = 1.9 \times 10^{-6}$; Deletion burden: Wilcoxon rank-sum test $P = 4.3 \times 10^{-2}$...”). In addition, the significant prognostic association of CNAB was also only observed in pCCA instead of iCCA with purity-adjusted results. We also added the results to Supplementary Table 6 and manuscript (with tracked changes:LINE 142; without tracked changes:LINE 139: “Consistent findings were obtained from the prognostic effect of CNAB based on purity-adjusted CNAB(pCCA: HR = 2.89, $P = 3.6 \times 10^{-2}$; iCCA: HR = 0.84,

$P = 4.5 \times 10^{-1}$)”). At last, we found that the purity-adjusted GISTIC2 results were consistent with the GATK protocol. All regions were identified as significant except six regions (3q29, 5p15.33 and 17q12 for amplification; 2p24.1, 3q29 and 19p12 for deletion). We added these results to Supplementary Table 8.

[**Comment5**] Do cases with *METTL14* mutations have LOH of the wild allele? It is unclear whether this mutation is heterozygous (maybe dominant negative) or two-hit inactivated (classical TSG).

[**Answer**] Thanks for the reviewer’s suggestion. We detected the copy number status of *METTL14* in the patient with mutation *METTL14* R298 from our xMU cohort using TitanCNA and found that *METTL14* is DIPLOID and it indicated that the role of *METTL14* is more likely to be dominant negative.

[**Comment6**] In Fig 4E/4F/4G, is the difference between NC and R298H significantly different? If R298H functions as a dominant-negative form, R298H should inhibit endogenous *METTL14* function. How about the *METTL14* mutation status in the two CCC cell lines?

[**Answer**] We thank the reviewer for their suggestion. In Fig 4E/4F/4G, there is no difference between NC and R298H significantly. Firstly, since the functional role of *METTL14*^{wt} in CCA has remained unclear, we revealed that *METTL14*^{wt} in CCA may act as a tumor suppressive effect by comparing the functional differences between NC and WT. To explore the functional role of *METTL14*^{R298H}, we further compared the functional differences between *METTL14*^{R298H} and *METTL14*^{wt} to demonstrated that *METTL14*^{R298H} mutation was sufficient to repressing the tumor suppressor function of *METTL14*^{wt} in CCA. Notably, the initial expression level of *METTL14* wild-type was low in the two cell lines used in our study. And the overexpression efficiency of *METTL14*^{R298H} transfected by our lentivirus was significant higher than the expression level of *METTL14*^{wt}. To identify *METTL14* mutation status in the two CCA cell lines, we used sanger sequencing to verified that there were no R298H mutations of *METTL14* in CCA cell lines (RBE and HCCC9810)(Supplementary Fig.3G).

[**Comment7**] Fig 4H and Fig 5G, there is a typo ("clony" should be colony)

[Answer] Thank the reviewer for pointing this out. We have revised this mistake in the revised manuscript.

[Comment8] How about the *METTL14* mutant effect on the *MET* transcript?

[Answer] Considering this comment of the reviewer, we added the qRT-PCR results to examine the *METTL14* mutant effect on the *MET* transcript in the mRNA level. And the result showed that *METTL14*^{wt} cells decreased the expression of *MET*, and no noticeable effect on *MET* expression was observed in *METTL14*^{R298H} cells (Figure.Response 3). At the same time, the functional role and mechanism of *MET* in the occurrence and development of malignant tumors are relatively clear. Our study aimed to reveal more about the role and mechanism of *MACF1* in CCA.

Figure.Response 3. *MET* mRNA expression in *METTL14*^{wt} and *METTL14*^{R298H} in CCA cell lines.

[Comment9] Do cases with low *METTL14* expression (Fig 4B) show higher *MACF1* in the primary tumor cohort?

[Answer] We thank the reviewer for this insightful suggestion. In our findings, we first validated *MACF1* upregulation in CCA using qRT-PCR, and noticed that lower *METTL14* mRNA level exhibited stronger *MACF1* expression in mRNA level (Fig.5L).

[Comment10] *METTL14* mutant effect on b-catenin signaling: In Fig 5K, the amount of nuclear b-catenin is comparable between NC and R298H expressing clones (lanes 4 and 6), but the expression of cyclinD1 is more decreased in R298H expressing clones. It seems that

METTL14-MACF1-WNT would be more complicated than the author proposed. How to settle this inconsistency?

[Answer] We thank the reviewer for this insightful suggestion. Subcellular fractionation (including nuclear, membrane, and cytoplasm) was performed using cell fractionation kit (Cell Signaling Technology). Equal amounts of total protein were separated by SDS-PAGE and subsequently transferred to nitrocellulose membrane. And the results showed that the increase of nuclear β -catenin was correlated with the expression of *METTL14*^{R298H} rather than *METTL14*^{wt} in both RBE and HCCC9810 cell lines. Notably, although we provide evidence to demonstrated that *METTL14*-mediated m⁶A modification repressed the *MACF1*/ β -catenin pathway in CCA, while *METTL14*^{R298H} mutation disrupted this mechanism, m⁶A-related key catalytic protein would be more complicated on the WNT pathway and the underlying mechanism is not fully characterized. Previously, ALKBH5 reducing m⁶A levels of WIF-1 and hindering activation of Wnt signaling². YTHDF bound METTL3-mediated m⁶A of APC mRNA and upregulated Wnt/ β -catenin pathway³. Given the importance of Wnt signaling in regulating cancer tumorigenesis, *METTL14-MACF1*-WNT would be more complicated, which deserves further investigation. As our ongoing study has found, *METTL14* and *METTL3* are also dependent on each other for m⁶A modification and expression correlation, which implied that m⁶A related key catalytic protein would apparently affect each other, underlying mechanism is not fully characterized. In view of this problem, we will add this problem as a limitation of this study in the revised manuscript, and we hope to explore this problem in future research (with tracked changes: Line 358-366; without tracked changes: Line 349-357).

Reference

2. Tang, B. *et al.* m(6)A demethylase ALKBH5 inhibits pancreatic cancer tumorigenesis by decreasing WIF-1 RNA methylation and mediating Wnt signaling. *Mol Cancer* **19**, 3 (2020).
3. Wang, W. *et al.* METTL3 promotes tumour development by decreasing APC expression mediated by APC mRNA N(6)-methyladenosine-dependent YTHDF binding. *Nat Commun* **12**, 3803 (2021).

REVIEWER COMMENTS

Reviewer #1 (Remarks to the Author):

The authors addressed Most of my comments satisfactorily. However, comment #6 (Reviewer #1) were not addressed. Supplementary Fig. 4C seems to be a summary/conclusion of the seq findings, not the original data. The original sequencing data and validation data need to be shown.

Reviewer #2 (Remarks to the Author):

The authors have nicely addressed my prior, minor concerns.

Reviewer #3 (Remarks to the Author):

Generally, the authors responded to comments raised by reviewer #3. I have no further issues to be revised.

Dear reviewers

We deeply appreciate for your positive and constructive comments and suggestions on our manuscript entitled “Genomic landscapes reveal post-transcriptional modifier disruption in cholangiocarcinoma”. We have made corresponding revisions to the manuscript as required. We would like to express our great appreciation to your kindness and patience.

Point-by-point responses to reviewer concerns:

Response to Reviewer #1:

[**Comment1**] The authors addressed Most of my comments satisfactorily. However, comment #6 (Reviewer #1) were not addressed. Supplementary Fig. 4C seems to be a summary/conclusion of the seq findings, not the original data. The original sequencing data and validation data need to be shown.

[**Answer**] We are grateful for the suggestion. To be clearer and in accordance with the reviewer's concerns, we have presented the information of m6A peaks across the *MACF1* transcript (Peak 1 and Peak 2) in Supplementary Fig.4C in the manuscript by referring to the example provided by Reviewer #1 (<https://www.nature.com/articles/nprot.2012.148>). To prove that *METTL14* targets *MACF1* mRNA for m6A modification, we validated the m6A-Seq data by MeRIP-qPCR. The results showed that *METTL14*^{R298H} significantly decreased the amount of *MACF1* modified by m⁶A compared to *METTL14*^{wt} (Figure Reponse.1).

Supplementary Fig.4C Gene plots of *MACF1* coding region harboring m6A peaks. Coverage of IP and input control is indicated in red and blue, respectively. The blue boxes in the bottom panel represent exons and UTRs.

Figure Reponse.1 The m⁶A modification level of *MACF1* in RBE and HCCC9810 cells was validated in methylated RNA Immunoprecipitation (MeRIP).

Response to Reviewer #2:

[Comment1] The authors have nicely addressed my prior, minor concerns.

[Answer] Our deepest gratitude goes to you for your careful work and thoughtful suggestions that have helped improve this paper substantially.

Response to Reviewer #3:

[Comment1] Generally, the authors responded to comments raised by reviewer #3. I have no further issues to be revised.

[Answer] Thank you for your careful review. We appreciate your efforts in reviewing our manuscript during this unprecedented and challenging time.

REVIEWERS' COMMENTS

Reviewer #1 (Remarks to the Author):

The authors have addressed my comments.

Dear reviewers

We deeply appreciate for your positive and constructive comments and suggestions on our manuscript entitled “Genomic landscapes reveal post-transcriptional modifier disruption in cholangiocarcinoma”. We would like to express our great appreciation to your kindness and patience.

Point-to-point response to reviewer concerns:

Response to Reviewer #1:

[**Comment 1**] The authors have addressed my comments.

[**Answer**] Our deepest gratitude goes to you for your careful work and thoughtful suggestions that have helped improve this paper substantially. We appreciate your efforts in reviewing our manuscript during this unprecedented and challenging time.